# The Behavior and Convergence of Local Bayesian Optimization

**Kaiwen Wu**
University of Pennsylvania
kaiwenwu@seas.upenn.edu

**Kyurae Kim**
University of Pennsylvania
kyrkim@seas.upenn.edu

**Roman Garnett**
Washington University in St. Louis
garnett@wustl.edu

**Jacob R. Gardner**
University of Pennsylvania
jacobrg@seas.upenn.edu

## Abstract

A recent development in Bayesian optimization is the use of local optimization strategies, which can deliver strong empirical performance on high-dimensional problems compared to traditional global strategies. The "folk wisdom" in the literature is that the focus on local optimization sidesteps the curse of dimensionality; however, little is known concretely about the expected behavior or convergence of Bayesian local optimization routines. We first study the behavior of the local approach, and find that the statistics of individual local solutions of Gaussian process sample paths are surprisingly good compared to what we would expect to recover from global methods. We then present the first rigorous analysis of such a Bayesian local optimization algorithm recently proposed by Müller et al. (2021), and derive convergence rates in both the noiseless and noisy settings.

## 1   Introduction

Bayesian optimization (BO) [11] is among the most well studied and successful algorithms for solving black-box optimization problems, with wide ranging applications from hyperparameter tuning [23], reinforcement learning [19, 27], to recent applications in small molecule [18, 26, 7, 13, 12] and sequence design [25]. Despite its success, the curse of dimensionality has been a long-standing challenge for the framework. Cumulative regret for BO scales exponentially with the dimensionality of the search space, unless strong assumptions like additive structure [15] are met. Empirically, BO has historically struggled on optimization problems with more than a handful of dimensions.

Recently, local approaches to Bayesian optimization have empirically shown great promise in addressing high-dimensional optimization problems [10, 19, 20, 27, 17]. The approach is intuitively appealing: a typical claim is that a local optimum can be found relatively quickly, and exponential sample complexity is only necessary if one wishes to enumerate all local optima. However, these intuitions have not been fully formalized. While global Bayesian optimization is well studied under controlled assumptions [24, 3], little is known about the behavior or convergence properties of local Bayesian optimization in these same settings — the related literature focuses mostly on strong empirical results on real-world applications.

In this paper, we begin to close this gap by investigating the behavior of and proving convergence rates for local BO under the same well-studied assumptions commonly used to analyze global BO. We divide our study of the properties of local BO in to two questions:

**Q1**  Using common assumptions (*e.g.*, when the objective function is a sample path from a GP with known hyperparameters), how good are the local solutions that local BO converges to?

37th Conference on Neural Information Processing Systems (NeurIPS 2023).

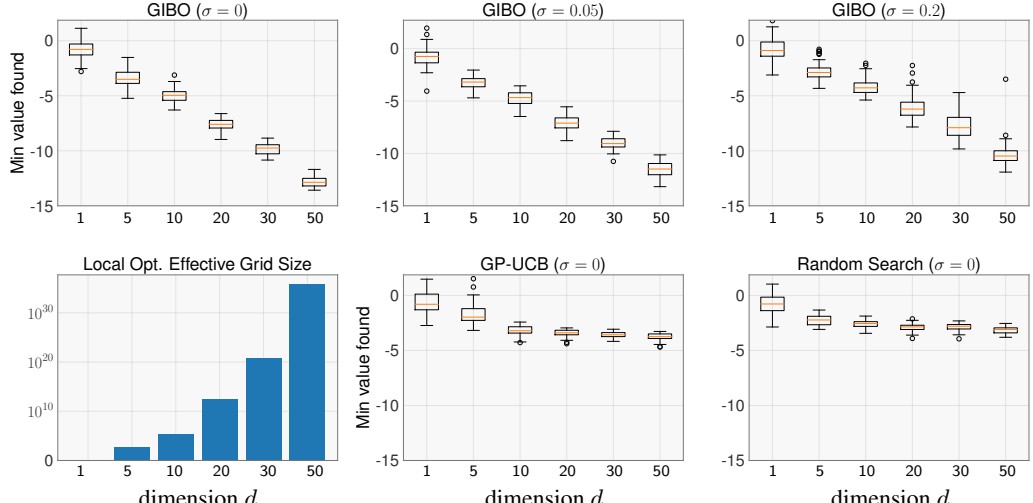

Figure 1: The unreasonable effectiveness of locally optimizing GP sample paths. **(Top row):** Distributions of local solutions found when locally optimizing GP sample paths in various numbers of dimensions, with varying amounts of noise. **(Bottom left):** The minimum sample complexity of grid search required to achieve the median value found by GIBO ($\sigma = 0$) in expectation. **(Bottom middle, right):** The performance of global optimization algorithms GP-UCB and random search in this setting. See §3 for details.

**Q2** Under these same assumptions, is there a local Bayesian optimization procedure that is guaranteed to find a local solution, and how quickly can it do so?

For **Q1**, we find empirically that the behavior of local BO on "typical" high dimensional Gaussian process sample paths is surprising: even a single run of local BO, without random restarts, results in shockingly good objective values in high dimension. Here, we may characterize "surprise" by the minimum equivalent grid search size necessary to achieve this value in expectation. See Figure 1 and more discussions in §3.

For **Q2**, we present the first theoretical convergence analysis of a local BO algorithm, GIBO as presented in [19] with slight modifications. Our results bring in to focus a number of natural questions about local Bayesian optimization, such as the impact of noise and the increasing challenges of optimization as local BO approaches a stationary point.

We summarize our contributions as follows:

1. We provide upper bounds on the uncertainty of the gradient when actively learned using the GIBO procedure described in [19] in both the noiseless and noisy settings.

2. By relating these uncertainty bounds to the bias in the gradient estimate used at each step, we translate these results into convergence rates for both noiseless and noisy functions.

3. We investigate the looseness in our bound by studying the empirical reduction in uncertainty achieved by using the GIBO policy and comparing it to our upper bound.

4. We empirically study the quality of individual local solutions found by a local Bayesian optimization, a slightly modified version of GIBO, on GP sample paths.

## 2 Background and Related Work

**Bayesian optimization.** Bayesian optimization (BO) considers minimizing a black-box function $f$:

$$\underset{\mathbf{x} \in \mathcal{X}}{\text{minimize}} \, f(\mathbf{x})$$

through potentially noisy function queries $y = f(\mathbf{x}) + \varepsilon$. When relevant to our discussion, we will assume an *i.i.d.* additive Gaussian noise model: $\varepsilon \sim \mathcal{N}(0, \sigma^2)$. To decide which inputs to

be evaluated, Bayesian optimization estimates $f$ via a *surrogate model,* which in turn informs a *policy* (usually realized by maximizing an *acquisition function* $\alpha(\mathbf{x})$ over the input space) to identify evaluation candidates. Selected candidates are then evaluated, and the surrogate model is updated with this new data, allowing the process to repeat with the updated surrogate model.

**Gaussian processes.** Gaussian processes (GP) are the most commonly used surrogate model class in BO. A Gaussian process $\mathcal{GP}(\mu, k)$ is fully specified by a mean function $\mu$ and a covariance function $k$. For a finite collection of inputs $\mathbf{X}$, the GP induces a joint Gaussian belief about the function values, $f(\mathbf{X}) \sim \mathcal{N}(\mu(\mathbf{X}), k(\mathbf{X}, \mathbf{X}))$. Standard rules about conditioning Gaussians can then be used to condition the GP on a dataset $\mathcal{D}$, resulting in an updated posterior process reflecting the information contained in these observations:

$$f_{\mathcal{D}} \sim \mathcal{GP}(\mu_{\mathcal{D}}, k_{\mathcal{D}}). \tag{1}$$

**Existing bounds for global BO.** Srinivas et al. [24] proved the first sublinear cumulative regret bounds of a global BO algorithm, `GP-UCB`, in the noisy setting. Their bounds have an exponential dependency on the dimension, a result that is generally unimproved without assumptions on the function structure like an additive decomposition [15]. This exponential dependence can be regarded as a consequence of the curse of dimensionality. In one dimension, Scarlett [21] has characterized the optimal regret bound (up to a logarithmic factor), and `GP-UCB` is near-optimal in the case of the RBF kernel. In the noiseless setting, improved convergence rates have been developed under additional assumptions. For example, De Freitas et al. [6] proved exponential convergence by assuming $f$ is locally quadratic in a neighborhood around the global optimum, and Kawaguchi et al. [16] proved an exponential simple regret bound under an additional assumption that the optimality gap $f(\mathbf{x}) - f^*$ is bounded by a semi-metric.

**Gaussian process derivatives.** A key fact about Gaussian process models that we will use is that they naturally give rise to gradient estimation. If $f$ is Gaussian process distributed, $f \sim \mathcal{N}(\mu, k)$, and $k$ is differentiable, then the gradient process $\nabla f$ also is also (jointly) distributed as a GP. Noisy observations $\mathbf{y}$ at arbitrary locations $\mathbf{X}$ and the gradient measured at an arbitrary location $\mathbf{x}$ are jointly distributed as:

$$\begin{pmatrix} \mathbf{y} \\ \nabla f(\mathbf{x}) \end{pmatrix} \sim \mathcal{N}\left( \begin{pmatrix} \mu(\mathbf{X}) \\ \nabla \mu(\mathbf{x}) \end{pmatrix}, \begin{pmatrix} k(\mathbf{X}, \mathbf{X}) + \sigma^2 \mathbf{I} & k(\mathbf{X}, \mathbf{x})\nabla^{\top} \\ \nabla k(\mathbf{x}, \mathbf{X}) & \nabla k(\mathbf{x}, \mathbf{x})\nabla^{\top} \end{pmatrix} \right).$$

This property allows probabilistic modeling of the gradient given noisy function observations. The gradient process conditioned on observations $\mathcal{D}$ is distributed as a Gaussian process:

$$\nabla f \mid \mathcal{D} \sim \mathcal{N}(\nabla \mu_{\mathcal{D}}, \nabla k_{\mathcal{D}} \nabla^{\top}).$$

## 3 How Good Are Local Solutions?

Several local Bayesian optimization algorithms have been proposed recently as an alternative to global Bayesian optimization due to their favourable empirical sample efficiency [10, 19, 20]. The common idea is to utilize the surrogate model to iteratively search for *local* improvements around some current input $\mathbf{x}_t$ at iteration $t$. For example, Eriksson et al. [10] centers a hyper-rectangular trust region on $\mathbf{x}_t$ (typically taken to be the best input evaluated so far), and searches locally within the trust region. Other methods like those of Müller et al. [19] or Nguyen et al. [20] improve the solution $\mathbf{x}_t$ locally by attempting to identify descent directions $\mathbf{d}_t$ so that $\mathbf{x}_t$ can be updated as $\mathbf{x}_{t+1} = \mathbf{x}_t + \eta_t \mathbf{d}_t$.

Before choosing a particular local BO method to study the convergence of formally, we begin by motivating the high level philosophy of local BO by investigating the quality of individual local solutions in a controlled setting. Specifically, we study the local solutions of functions $f$ drawn from Gaussian processes with known hyperparameters. In this setting, Gaussian process sample paths can be drawn as differentiable functions adapting the techniques described in Wilson et al. [29]. Note that this is substantially different from the experiment originally conducted in Müller et al. [19], where they condition a Gaussian process on 1000 examples and use the posterior mean as an objective.

To get a sense of scale for the optimum, we first analyze roughly how well one might expect a simple grid search baseline to do with varying sample budgets. A well known fact about the order statistics of Gaussian random variables is the following:

**Algorithm 1:** A Local Bayesian Optimization Algorithm

---

**Input:** A black-box function $f$ over a convex compact set $\mathcal{X}$.

**for** $t = 1, 2, \cdots, T$ **do**

    $\mathbf{X} = \mathrm{argmin}_{\mathbf{Z}}\, \alpha_{\mathrm{trace}}(\mathbf{x}_t, \mathbf{Z})$ where $\mathbf{Z} \in \mathbb{R}^{b_t \times d}$            `// b_t is the batch size`

    evaluate the black-box function $f$ on $\mathbf{X}$, obtaining (possibly noisy) measurements $\mathbf{y}$

    $\mathcal{D}_t = \mathcal{D}_{t-1} \cup (\mathbf{X}, \mathbf{y})$                           `// add (X, y) to the training data`

    $\mathbf{x}_{t+1} = \mathbf{x}_t - \eta_t \nabla \mu_{\mathcal{D}_t}(\mathbf{x}_t)$

**end**

---

**Remark 1** (*e.g.*, [8, 14]). *Let $X_1, X_2, \cdots, X_n$ be (possibly correlated) Gaussian random variables with marginal variance $s^2$, and let $Y = \max\{X_i\}$. Then the expected maximum is bounded by*

$$\mathbb{E}[Y] \leq s\sqrt{2 \log n}.$$

This result directly implies an upper bound on the expected maximum (or minimum by symmetry) of a grid search with $n$ samples. The logarithmic bound results in an optimistic estimate of the expected maximum (and the minimum): as $n \to \infty$ the bound goes to infinity as well, whereas the maximum (and the minimum) of a GP is almost surely finite due to the correlation in the covariance.

With this, we now turn to evaluating the quality of local solutions. To do this, we optimize 50 sample paths starting from $\mathbf{x} = \mathbf{0}$ from a centered Gaussian process with a RBF kernel (unit outputscale and unit lengthscale) using a variety of observation noise standard deviations $\sigma$. We then run iterations of local Bayesian optimization (as described later in Algorithm 1) to convergence or until an evaluation budget of 5000 is reached. In the noiseless setting ($\sigma = 0$), we modify Algorithm 1 to pass our gradient estimates to BFGS rather than applying the standard gradient update rule for efficiency.

We repeat this procedure for $d \in \{1, 5, 10, 20, 30, 50\}$ dimensions and $\sigma \in \{0, 0.05, 0.2\}$ and display results in Figure 1. For each dimension, we plot the distribution of single local solutions found from each of the 50 trials as a box plot. In the noiseless setting, we find that by $d = 50$, a single run of local optimization (*i.e.*, without random restarts) is able to find a median objective value of $-12.9$, corresponding to a grid size of at least $n = 10^{36}$ to achieve this value in expectation!

For completeness, we provide results of running `GP-UCB` and random search also with budgets of 5000 and confirm the effectiveness of local optimization in high dimensions.

## 4 A Local Bayesian Optimization Algorithm

The study above suggests that, under assumptions commonly used to theoretically study the performance of Bayesian optimization, finding even a single local solution is surprisingly effective. The natural next question is whether and how fast we can find them. To understand the convergence of local BO, we describe a prototypical local Bayesian optimization algorithm that we will analyze in Algorithm 1, which is nearly identical to `GIBO` as described in [19] with one exception: Algorithm 1 allows varying batch sizes $b_t$ in different iterations.

Define $k_{\mathcal{D} \cup \mathbf{Z}}(\mathbf{x}_t, \mathbf{x}_t) = k_{\mathcal{D}}(\mathbf{x}_t, \mathbf{x}_t) - k_{\mathcal{D}}(\mathbf{x}_t, \mathbf{Z})(k_{\mathcal{D}}(\mathbf{Z}, \mathbf{Z}) + \sigma^2 \mathbf{I})^{-1} k_{\mathcal{D}}(\mathbf{Z}, \mathbf{x}_t)$. Namely, $k_{\mathcal{D} \cup \mathbf{Z}}(\cdot, \cdot)$ is the posterior covariance function conditioned on the data $\mathcal{D}$ as well as new inputs $\mathbf{Z}$. Crucially, observe that the posterior covariance of a GP does not depends on the labels $\mathbf{y}$, hence the compact notation $k_{\mathcal{D} \cup \mathbf{Z}}$. The acquisition function $\alpha_{\mathrm{trace}}$ of `GIBO` is defined as

$$\alpha_{\mathrm{trace}}(\mathbf{x}, \mathbf{Z}) = \mathrm{tr}\big(\nabla k_{\mathcal{D} \cup \mathbf{Z}}(\mathbf{x}_t, \mathbf{x}_t) \nabla^\top\big), \tag{2}$$

which is the trace of the posterior gradient covariance conditioned on the union of the training data $\mathcal{D}$ and candidate designs $\mathbf{Z}$.

The posterior gradient covariance $\nabla k_{\mathcal{D}}(\mathbf{x}_t, \mathbf{x}_t) \nabla^\top$ quantifies the uncertainty in the gradient $\nabla f(\mathbf{x}_t)$. By minimizing the acquisition function $\alpha_{\mathrm{trace}}(\mathbf{x}_t, \mathbf{Z})$ over $\mathbf{Z}$, we actively search for designs $\mathbf{Z}$ that minimizes the one-step look-ahead uncertainty, where the uncertainty is measured by the trace of the posterior covariance matrix. After selecting a set of locations $\mathbf{Z}$, the algorithm queries the (noisy) function values at $\mathbf{Z}$, and updates the dataset. The algorithm then follows the (negative) expected gradient $\nabla \mu_{\mathcal{D}_t}(\mathbf{x}_t)$ to improve the current iterate $\mathbf{x}_t$, and the process repeats.

# 5 Convergence Results

In light of the strong motivation we now have for local Bayesian optimization, we will now establish convergence guarantees for the simple local Bayesian optimization routine outlined in the previous section, in both the noiseless and noisy setting. We first pause to establish some notations and a few assumptions that will be used throughout our results.

**Notation.** Let $\mathcal{X} \subset \mathbb{R}^d$ be a convex compact domain (*e.g.*, $\mathcal{X} = [0,1]^d$). Let $\mathcal{H}$ be a reproducing kernel Hilbert space (RKHS) on $\mathcal{X}$ equipped with a reproducing kernel $k(\cdot, \cdot)$. We use $\|\cdot\|$ to denote the Euclidean norm and $\|\cdot\|_{\mathcal{H}}$ to denote the RKHS norm. The minimum of a continuous function $f$ defined on a compact domain $\mathcal{X}$ is denoted as $f^* = \min_{\mathbf{x} \in \mathcal{X}} f(\mathbf{x})$.

In our convergence results, we make the following assumption about the kernel $k$. Many commonly used kernels satisfy this assumption, *e.g.*, the RBF kernel and Matérn kernel with $\nu > 2$.

**Assumption 1.** *The kernel $k$ is stationary, four times continuously differentiable and positive definite.*

The following is a technical assumption that simplifies proofs and convergence rates.

**Assumption 2.** *The iterates $\{\mathbf{x}_t\}_{t=1}^{\infty}$ stay in the interior of the domain $\mathcal{X}$.*

Intuitively, with the domain $\mathcal{X}$ large enough and an appropriate initialization, the iterates most likely always stay in the interior. In the case when the iterate $\mathbf{x}_{t+1}$ slides out of the domain $\mathcal{X}$, we can add a projection operator to project it back to $\mathcal{X}$, which will be discussed in §5.2.

We also define a particular (standard) smoothness measure that will be useful for proving convergence.

**Definition 1** (Smoothness). *A function $f$ is $L$-smooth if and only if for all $\mathbf{x}_1, \mathbf{x}_2 \in \mathcal{X}$, we have:*
$$\|\nabla f(\mathbf{x}_1) - \nabla f(\mathbf{x}_2)\| \leq L\|\mathbf{x}_1 - \mathbf{x}_2\|.$$

Next, we define the notion of an *error function*, which quantifies the maximum achievable reduction in uncertainty about the gradient at $\mathbf{x} = 0$ using a set of $b$ points $\mathbf{Z}$ and no other data, *i.e.*, the data $\mathcal{D}$ is an empty set in the acquisition function (2).

**Definition 2** (Error function). *Given input dimensionality d, kernel k and noise standard deviation $\sigma$, we define the following error function:*
$$E_{d,k,\sigma}(b) = \inf_{\mathbf{Z} \in \mathbb{R}^{b \times d}} \mathrm{tr}\big(\nabla k(\mathbf{0}, \mathbf{0})\nabla^{\top} - \nabla k(\mathbf{0}, \mathbf{Z})(k(\mathbf{Z}, \mathbf{Z}) + \sigma^2 \mathbf{I})^{-1} k(\mathbf{Z}, \mathbf{0})\nabla^{\top}\big). \tag{3}$$

## 5.1 Convergence in the Noiseless Setting

We first prove the convergence of Algorithm 1 with noiseless observations, *i.e.*, $\sigma = 0$. Where necessary, the inverse kernel matrix $k_{\mathcal{D}}(\mathbf{Z}, \mathbf{Z})^{-1}$ should be interpreted as the pseudoinverse. In this section, we will assume the ground truth function $f$ is a function in the RKHS with bounded norm. This assumption is standard in the literature [*e.g.*, 3, 24], and the results presented here ultimately extend trivially to the other assumption commonly made (that $f$ is a GP sample path).

**Assumption 3.** *The ground truth function $f$ is in $\mathcal{H}$ with bounded norm $\|f\|_{\mathcal{H}} \leq B$.*

Because $k$ is four times continuously differentiable, $f$ is twice continuously differentiable. Furthermore, on a compact domain $\mathcal{X}$, $f$ is guaranteed to be $L$-smooth for some constant $L$ (see Lemma 9 for a complete explanation). To prove the convergence of Algorithm 1, we will show the posterior mean gradient $\nabla \mu_{\mathcal{D}}$ approximates the ground truth gradient $\nabla f$. By using techniques in meshless data approximation [*e.g.*, 28, 5], we prove the following error bound, which precisely relates the gradient approximation error and the posterior covariance trace.

**Lemma 1.** *For any $f \in \mathcal{H}$, any $\mathbf{x} \in \mathcal{X}$ and any $\mathcal{D}$, we have the following inequality*
$$\|\nabla f(\mathbf{x}) - \nabla \mu_{\mathcal{D}}(\mathbf{x})\|^2 \leq \mathrm{tr}\big(\nabla k_{\mathcal{D}}(\mathbf{x}, \mathbf{x})\nabla^{\top}\big)\|f\|_{\mathcal{H}}^2. \tag{4}$$

Since $f$ has bounded norm $\|f\|_{\mathcal{H}} \leq B$, the right hand side of (4) is a multiple of the posterior covariance trace, which resembles the acquisition function (2). Indeed, the acquisition function (2) can be interpreted as minimizing the one-step look-ahead worst-case gradient estimation error in the RKHS $\mathcal{H}$, which justifies GIBO from a different perspective.

Next, we provide a characterization of the error function $E_{d,k,0}$ under the noiseless assumption $\sigma = 0$. This characterization will be useful later to express the convergence rate.

**Lemma 2.** *For $\sigma = 0$, the error function is bounded by $E_{d,k,0}(b) \leq C \max\{0, 1 + d - b\}$, where $C = \max_{1 \leq i \leq d} \frac{\partial^2}{\partial x_i \partial x_i'} k(\mathbf{0}, \mathbf{0})$ is the maximum of the Hessian's diagonal entries at the origin.*

Now we are ready to present the convergence rate of Algorithm 1 under noiseless observations.

**Theorem 1.** *Let $f \in \mathcal{H}$ whose smoothness constant is L. Running Algorithm 1 with constant batch size $b_t = b$ and step size $\eta_t = \frac{1}{L}$ for T iterations outputs a sequence satisfying*

$$\min_{1 \leq t \leq T} \|\nabla f(\mathbf{x}_t)\|^2 \leq \frac{1}{T}\big(2L(f(\mathbf{x}_1) - f^*)\big) + B^2 \cdot E_{d,k,0}(b). \tag{5}$$

As $T \to \infty$, the first term in the right hand side of (5) decays to zero, but the second term may not. Thus, with constant batch size $b$, Algorithm 1 converges to a region where the squared gradient norm is upper bounded by $B^2 E_{d,k,0}(b)$ with convergence speed $\mathcal{O}(\frac{1}{T})$. An important special case occurs when the batch size $b_t = d + 1$ in every iteration. In this case, $E_{d,k,0}(b) = 0$ by Lemma 2 and the algorithm converges to a stationary point with rate $\mathcal{O}(\frac{1}{T})$.

**Corollary 1.** *Under the same assumptions of Theorem 1, using batch size $b_t = d + 1$, we have*

$$\min_{1 \leq t \leq T} \|\nabla f(\mathbf{x}_t)\|^2 \leq \frac{1}{T}\big(2L(f(\mathbf{x}_1) - f^*)\big).$$

*Therefore, the total number of samples $n = \mathcal{O}(dT)$ and the squared gradient norm $\|\nabla f(\mathbf{x}_t)\|^2$ converges to zero at the rate $\mathcal{O}(d/n)$.*

We highlight that the rate is linear with respect to the input dimension $d$. Thus, as expected, local Bayesian optimization finds a stationary point in the noiseless setting with significantly better non-asymptotic rates in $d$ than global Bayesian optimization finds the global optimum. Of course, the fast convergence rate of Algorithm 1 is at the notable cost of not finding the global minimum. However, as Figure 1 demonstrates and as discussed in §3, a single stationary point may already do very well under the assumptions used in both analyses.

We note that the results in this section can be extended to the GP sample path assumption as well. Due to space constraint, we defer the result to Theorem 4 in the appendix.

## 5.2 Convergence in the Noisy Setting

We now turn our attention to the more challenging and interesting setting where the observation noise $\sigma > 0$. We analyze convergence under the following GP sample path assumption, which is commonly used in the literature [*e.g.*, 24, 6]:

**Assumption 4.** *The objective function $f$ is a sample from a GP, $f \sim \mathcal{GP}(\mu, k)$ with known mean function, covariance function, and hyperparameters. Observations of the objective function $y$ are made with added Gaussian noise of known variance, $y = f(\mathbf{x}) + \varepsilon$, where $\varepsilon \sim \mathcal{N}(0, \sigma^2)$.*

Since the kernel $k$ is four times continuously differentiable, $f$ is almost surely smooth.

**Lemma 3.** *For $0 < \delta < 1$, there exists a constant $L > 0$ such that $f$ is L-smooth w.p. at least $1 - \delta$.*

**Challenges.**  The observation noise introduces two important challenges. First, by the GP sample path assumption, we have:

$$\nabla f(\mathbf{x}_t) \sim \mathcal{N}(\nabla \mu_{\mathcal{D}_t}(\mathbf{x}_t), \nabla k_{\mathcal{D}_t}(\mathbf{x}_t, \mathbf{x}_t)\nabla^\top).$$

Thus, the posterior mean gradient $\nabla \mu_{\mathcal{D}_t}(\mathbf{x}_t)$ is an approximation of the ground truth gradient $\nabla f(\mathbf{x}_t)$, where the approximation error is quantified by the posterior covariance $\nabla k_{\mathcal{D}_t}(\mathbf{x}_t, \mathbf{x}_t)\nabla^\top$. For any $f$, we emphasize that the posterior mean gradient $\nabla \mu_{\mathcal{D}_t}(\mathbf{x}_t)$ is therefore *biased* whenever the posterior covariance is nonzero. Unfortunately, because of noise $\sigma > 0$, this is always true for any finite data $\mathcal{D}_t$. Thus, the standard analysis of stochastic gradient descent does not apply, as it typically assumes the stochastic gradient is *unbiased*.

The second challenge is that the noise directly makes numerical gradient estimation difficult. To build intuition, consider a finite differencing rule that approximates the partial derivative

$$\frac{\partial f}{\partial x_i} = \frac{1}{2h}\big[f(\mathbf{x} + h\mathbf{e}_i) - f(\mathbf{x} - h\mathbf{e}_i)\big] + \mathcal{O}(h^2)$$

where $\mathbf{e}_i$ is the $i$-th standard unit vector. In order to reduce the $\mathcal{O}(h^2)$ gradient estimation error, we need $h \to 0$. However, the same estimator under the noisy setting

$$\frac{\partial f}{\partial x_i} \approx \frac{1}{2h}\big[\big(f(\mathbf{x} + h\mathbf{e}_i) + \epsilon_1\big) - \big(f(\mathbf{x} - h\mathbf{e}_i) + \epsilon_2\big)\big] + \mathcal{O}(h^2),$$

where $\epsilon_1, \epsilon_2$ are *i.i.d.* Gaussians, diverges to infinity when $h \to 0$. Note that the above estimator is biased even with repeated function evaluations because $h$ cannot go to zero.

We first present a general convergence rate for Algorithm 1 that bounds the gradient norm.

**Theorem 2.** *For $0 < \delta < 1$, suppose $f$ is a GP sample whose smoothness constant is $L$ w.p. at least $1 - \delta$. Algorithm 1 with batch size $b_t$ and step size $\eta_t = \frac{1}{L}$ produces a sequence satisfying*

$$\min_{1 \le t \le T}\|\nabla f(\mathbf{x}_t)\|^2 \le \tfrac{1}{T}\big(2L(f(\mathbf{x}_1) - f^*)\big) + \tfrac{1}{T}\sum_{t=1}^{T} C_t E_{d,k,\sigma}(b_t) \tag{6}$$

*with probability at least $1 - 2\delta$, where $C_t = 2\log\big((\pi^2/6)(t^2/\delta)\big)$.*

The second term $\frac{1}{T}\sum_{t=1}^{T} C_t E_{d,k,\sigma}(b_t)$ in the right hand side of (6) is the average cumulative bias of the gradient. To finish the convergence analysis, we must further bound the error function $E_{d,k,\sigma}(b_t)$. For the RBF kernel, we obtain the following bound:

**Lemma 4** (RBF Kernel). *Let $k(\mathbf{x}_1, \mathbf{x}_2) = \exp\big(-\frac{1}{2}\|\mathbf{x}_1 - \mathbf{x}_2\|^2\big)$ be the RBF kernel. We have*

$$E_{d,k,\sigma}(2md) \le d\left(1 + W\left(-\frac{m}{e(m + \sigma^2)}\right)\right) = \mathcal{O}(\sigma d m^{-\frac{1}{2}}),$$

*where $m \in \mathbb{N}$ and $W$ denotes the principal branch of the Lambert W function.*

The error function (3) is an infimum over all possible designs, which is intractable for analysis. Instead, we analyze the infimum over a subset of designs of particular patterns (based on finite differencing), which can be solved analytically, resulting the first inequality. The second equality is proved by further bounding the Lambert function by its Taylor expansion at $-1/e$.

In addition, we obtain a similar bound for the Matérn kernel with $\nu = \frac{5}{2}$ by a slightly different proof.

**Lemma 5** (Matern Kernel). *Let $k(\cdot, \cdot)$ be the $\nu = 2.5$ Matérn kernel. Then, we have*

$$E_{d,k,\sigma}(2md) \lesssim \sigma d m^{-\frac{1}{2}} + \sigma^{\frac{3}{2}} d m^{-\frac{3}{4}} = \mathcal{O}(\sigma d m^{-\frac{1}{2}}).$$

Interestingly, Lemma 4 and Lemma 5 end up with the same asymptotic rate. Writing the bound in terms of the batch size $b$, we can see that $E_{d,k,\sigma}(b) = \mathcal{O}(\sigma d^{\frac{3}{2}} b^{-\frac{1}{2}})$ for both the RBF kernel and the $\nu = 2.5$ Matérn kernel.[1] Coupled with Theorem 2, the above lemmas translate into the following convergence rates, depending on the batch size $b_t$ in each iteration:

**Corollary 2.** *Let $k(\cdot, \cdot)$ be either the RBF kernel or the $\nu = 2.5$ Matérn kernel. Under the same conditions as Theorem 2, if*

$$b_t = \begin{cases} d\log^2 t; \\ dt; \\ dt^2, \end{cases} \quad then \quad \min_{1 \le t \le T}\|\nabla f(\mathbf{x}_t)\|^2 = \begin{cases} \mathcal{O}(1/T) + \mathcal{O}(\sigma d); \\ \mathcal{O}\big(\sigma d T^{-\frac{1}{2}}\log T\big) = \mathcal{O}\big(\sigma d^{\frac{5}{4}} n^{-\frac{1}{4}}\log n\big); \\ \mathcal{O}\big(\sigma d T^{-1}\log^2 T\big) = \mathcal{O}\big(\sigma d^{\frac{4}{3}} n^{-\frac{1}{3}}\log^2 n\big), \end{cases}$$

*with probability at least $1 - 2\delta$. Here, $T$ is the total number of iterations and $n$ is the total number of samples queried.*

---

[1] We suspect a similar bound holds for the entire Matérn family.

With nearly constant batch size $b_t = d\log^2 t$, Algorithm 1 converges to a region where the squared gradient norm is $\mathcal{O}(\sigma d)$. With linearly increasing batch size, the algorithm converges to a stationary point with rate $\mathcal{O}(\sigma d^{1.25}n^{-0.25}\log n)$, significantly slower than the $\mathcal{O}(d/n)$ rate in the noiseless setting. The quadratic batch size is nearly optimal up to a logarithm factor — increasing the batch size further slows down the rate (see Appendix D for details).

To achieve convergence to a stationary point using Corollary 2, the batch size $b_t$ must increase as optimization progresses. We note this may not be an artifact of our theory or our specific realization of local BO, but rather a general fact that is likely to be *implicitly* true for any local BO routine. For example, a practical implementation of Algorithm 1 might use a constant batch size and a line search subroutine where the iterate $\mathbf{x}_t$ is updated only when the (noisy) objective value decreases — otherwise, the iterate $\mathbf{x}_t$ does not change and the algorithm queries more candidates to reduce the bias in the gradient estimate. With this implementation, the batch size is increased repeatedly on any iterate while the line search condition fails. As the algorithm converges towards a stationary point, the norm of the ground-truth gradient $\|\nabla f(\mathbf{x}_t)\|$ decreases and thus requires more accurate gradient estimates. Therefore, the line search condition may fail more frequently with the constant batch size, and the effective batch size increases implicitly.

We provide two additional remarks on convergence in the noisy setting. First, the convergence rate is significantly slower than in the noiseless setting, highlighting the difficulty presented by noise. Second, when the noise is small, the convergence rate is faster. If $\sigma \to 0$, the rate is dominated by a lower-order term in the big $\mathcal{O}$ notation, recovering the $\mathcal{O}(\frac{1}{T})$ rate in the noiseless setting (see Appendix D for details). This is in sharp contrast with the existing analysis of BO algorithms. Existing convergence proofs in the noisy setting rely on analyzing the information gain, which is vacuous when $\sigma \to 0$, requiring new tools. It is interesting that no new tools are required here.

Finally, we revisit Assumption 2. In the case when it does not hold, we use a modified update:

$$\mathbf{x}_{t+1} = \mathrm{proj}_{\mathcal{X}}\big(\mathbf{x}_t - \eta_t \nabla\mu_{\mathcal{D}_t}(\mathbf{x}_t)\big), \tag{7}$$

where the projection operator $\mathrm{proj}_{\mathcal{X}}(\cdot)$ is defined as $\mathrm{proj}_{\mathcal{X}}(\mathbf{x}) = \mathrm{argmin}_{\mathbf{z}\in\mathcal{X}}\|\mathbf{z}-\mathbf{x}\|$, *i.e.*, the closest feasible point to $\mathbf{x}$. When the iterates stay in the interior of the domain $\mathcal{X}$, the projection operator is an identity map and the update rule simply reduces to $\mathbf{x}_{t+1} = \mathbf{x}_t - \eta_t \nabla\mu_{\mathcal{D}_t}(\mathbf{x}_t)$.

Define the gradient mapping

$$G(\mathbf{x}_t) = \tfrac{1}{\eta_t}\big(\mathbf{x}_t - \mathrm{proj}_{\mathcal{X}}(\mathbf{x}_t - \eta_t\nabla f(\mathbf{x}_t))\big),$$

which is a generalization of the gradient in the constrained setting: when $\mathbf{x}_t - \eta_t\nabla f(\mathbf{x}_t)$ lies in the interior of $\mathcal{X}$, the gradient mapping reduces to the (usual) gradient $\nabla f(\mathbf{x}_t)$. The following gives convergence rates of $\|G(\mathbf{x}_t)\|$.

**Theorem 3.** *Under the same conditions as Corollary 2, without Assumption 2, using the projected update rule* (7)*, Algorithm 1 obtains the following rates:*

$$\text{if}\quad b_t = \begin{cases} dt; \\ dt^2, \end{cases} \quad\text{then}\quad \min_{1\le t\le T}\|G(\mathbf{x}_t)\|^2 = \begin{cases} \mathcal{O}\big(\sigma d^{\frac{5}{4}}n^{-\frac{1}{4}}\log n + \sigma^{\frac{1}{2}}d^{\frac{5}{8}}n^{-\frac{1}{8}}\log n\big); \\ \mathcal{O}\big(\sigma d^{\frac{4}{3}}n^{-\frac{1}{3}}\log^2 n + \sigma^{\frac{1}{2}}d^{\frac{2}{3}}n^{-\frac{1}{6}}\log n\big), \end{cases}$$

*with probability at least $1 - 2\delta$. Here, $n$ is the total number of samples queried.*

These rates are slower than Corollary 2. We defer more details to Appendix C.

## 6 Additional Experiments

In this section, we investigate numerically the bounds in our proofs and study situations where the assumptions are violated. We focus on analytical experiments, because the excellent empirical performance of local BO methods on high dimensional real world problems has been well established in prior work [*e.g.*, 10, 19, 20, 27]. Detailed settings and additional experiments are available in Appendix E. The code is available at `https://github.com/kayween/local-bo-convergence`.

### 6.1 How loose are our convergence rates?

This section investigates the tightness of the bounds on the error function $E_{d,k,\sigma}(b)$ — a key quantity in our convergence rates. We plot in Figure 2 the error function $E_{d,k,\sigma}$ for the $\nu = 2.5$ Matérn

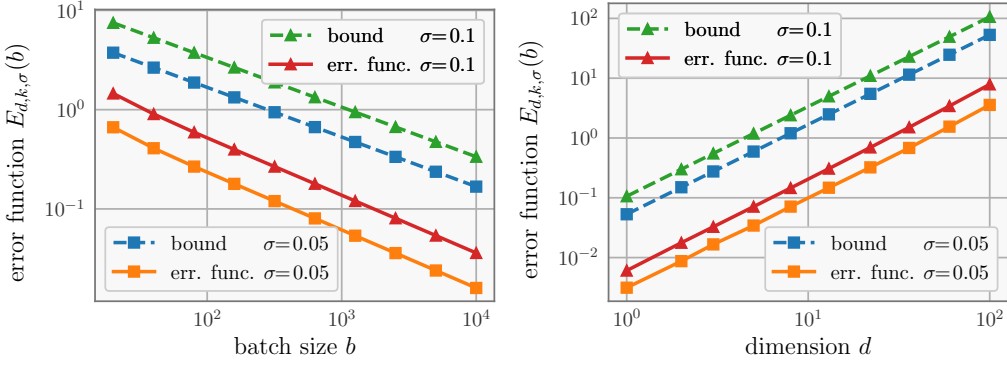

(a) fixed dimension, varying batch sizes      (b) fixed batch size, varying dimensions

Figure 2: Compare the error function (3) of the $\nu = 2.5$ Matérn kernel and our upper bound in Lemma 5. The error function $E_{d,k,\sigma}(b)$ is approximated by minimizing (3) with L-BFGS. Both plots are in log-log scale. **Left:** The slope indicates the exponent on $b$. Since the slope magnitude of the error function is slightly larger, the error function might decreases slightly faster than $\mathcal{O}(b^{-\frac{1}{2}})$ asymptotically. **Right:** The slope indicates the exponent on $d$. Since all lines have roughly the same slope, the dependency on the dimension in Lemma 5 seems to be tight.

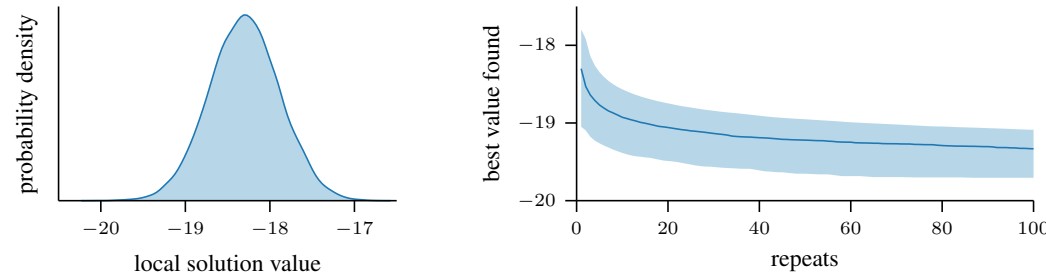

Figure 3: The performance of random restart on a GP sample path in 100 dimensions. **Left**: a density plot for the minimum value found on a single restart (compare with Figure 1). **Right**: the median and a 90% confidence interval for the best value found after a given number of random restarts.

kernel and our bound $\mathcal{O}(\sigma d^{\frac{3}{2}} b^{-\frac{1}{2}})$ implied by Lemma 5. The error function is empirically estimated by minimizing the trace of posterior gradient covariance (3) using L-BFGS. Figure 2a uses a fixed dimension $d = 10$ and varying batch sizes, while Figure 2b uses a fixed batch size $b = 500$ and varying dimensions. Both plots are in log-log scale. Therefore, the slope corresponds to the exponents on $b$ and $d$ in the bound $\mathcal{O}(\sigma d^{\frac{3}{2}} b^{-\frac{1}{2}})$, and the vertical intercept corresponds to the constant factor in the big $\mathcal{O}$ notation.

Figure 2a shows that the actual decay rate of the error function $E_{d,k,\sigma}(b)$ may be slightly faster than $\mathcal{O}(b^{-\frac{1}{2}})$, as the slopes of the (empirical) error function have magnitude slightly larger than $\frac{1}{2}$. Interestingly, Figure 2b demonstrates that the dependency $d^{\frac{3}{2}}$ on the dimension is quite tight — all lines in this plot share a similar slope magnitude.

## 6.2 What is the effect of multiple restarts?

In §3, we analyze local solutions found by a single run on different GP sample paths. Here, we investigate the impact of performing *multiple* restarts on the same sample path. In Figure 3, we plot (left) a kernel density estimate of the local solution found for a series of 10 000 random restarts, and (right) the best value found after several restarts with a 90% confidence interval. We make two observations. First, the improvement of running 10–20 restarts over a single restart is still significant: the Gaussian tails involved here render a difference of $\pm 1$ in objective value relatively large. Second, the improvement of multiple restarts saturates relatively quickly. This matches empirical observations

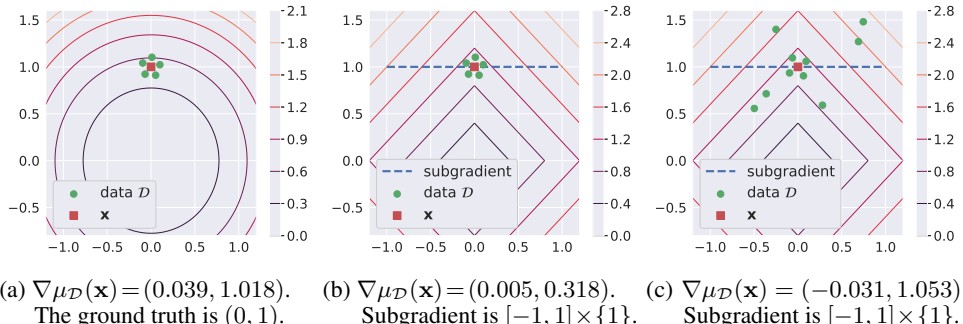

(a) $\nabla\mu_{\mathcal{D}}(\mathbf{x})=(0.039, 1.018)$.   (b) $\nabla\mu_{\mathcal{D}}(\mathbf{x})=(0.005, 0.318)$.   (c) $\nabla\mu_{\mathcal{D}}(\mathbf{x}) = (-0.031, 1.053)$.
    The ground truth is $(0, 1)$.          Subgradient is $[-1, 1]\times\{1\}$.          Subgradient is $[-1, 1]\times\{1\}$.

Figure 4: Estimating the "derivative" at $\mathbf{x} = (0, 1)$ with a Matérn Gausssian process ($\nu = 2.5$) in three different settings. **Left:** $f(\mathbf{x}) = \frac{1}{2}\|\mathbf{x}\|^2$. With $n = 5$ samples, the posterior mean gradient is close to the ground truth. **Middle:** $f(\mathbf{x}) = \|\mathbf{x}\|_1$. The $\ell_1$ norm is not differentiable at $(0, 1)$. With exactly the same samples as the left panel, the posterior mean gradient has higher error. **Right:** $f(\mathbf{x}) = \|\mathbf{x}\|_1$. Increasing the sample size to $n = 10$ decreases the estimation error.

made when using local BO methods on real world problems, where optimization is often terminated after (at most) a handful of restarts.

### 6.3 What if the objective function is non-differentiable?

In this section, we investigate what happens when $f$ is not differentiable — a setting ruled out in our theory by Assumption 1. In this case, what does the posterior mean gradient $\nabla\mu_{\mathcal{D}}$ learn from the oracle queries? To answer this question, we consider the $\ell_1$ norm function $\mathbf{x} \mapsto \|\mathbf{x}\|_1$ in two dimensions, which is non-differentiable when either $x_1$ or $x_2$ is zero. The $\ell_1$ norm is convex and thus its subdifferential is well-defined:

$$\partial\|\mathbf{x}\|_1 = \{\mathbf{g} \in \mathbb{R}^2 : \|\mathbf{g}\|_\infty \leq 1, \mathbf{g}^\top\mathbf{x} = \|\mathbf{x}\|_1\}.$$

In Figure 4, we use the posterior mean gradient $\nabla\mu_{\mathcal{D}}$ to estimate the (sub)gradient at $(0, 1)$. As a comparison, we also show the result for the differentiable quadratic function $\mathbf{x} \mapsto \frac{1}{2}\|\mathbf{x}\|^2$. Comparing Figure 4a and 4b, we observe that the $\ell_1$ norm has higher estimation error than the Euclidean norm despite using exactly the same set of queries. This might suggest that non-differentiability increases the sample complexity of learning the "slope" of the function. Increasing the sample size to $n = 10$ eventually results in a $\nabla\mu_{\mathcal{D}}(\mathbf{x})$ much closer to the subgradient, as shown in Figure 4c.

## 7 Discussion and Open Questions

For prototypical functions such as GP sample paths with RBF kernel, the local solutions discovered by local Bayesian optimization routines in high dimension are of surprisingly high quality. This motivates the theoretical study of these routines, which to date has been somewhat neglected. Here we have established the convergence of a recently proposed local BO algorithm, GIBO, in both the noiseless and noisy settings. The convergence rates are polynomial in *both* the number of samples and the input dimension, supporting the obvious intuitive observation that finding local optima is easier than finding the global optimum.

Our developments in this work leave a number of open questions. It is not clear whether solutions from local optimization perform so well because the landscape is inherently easy (*e.g.*, most stationary points are good approximation of the global minimum) or because the local BO has an unknown algorithmic bias that helps it avoid bad stationary points. This question calls for analysis of the landscape of GP sample paths (or the RKHS). Additionally, we conjecture that our convergence rates are not yet tight; §6.1 suggests that there is likely room for improvement. Further, it would be interesting to establish convergence for trust-region based local BO [*e.g.* 10, 9].

In practice, while existing lower bounds imply that any algorithm seeking to use local BO as a subroutine to discover the global optimum will ultimately face the same exponential-in-$d$ sample complexities as other methods, our results strongly suggest that, indeed as explored empirically in the literature, local solutions can not only be found, but can be surprisingly competitive.

## Acknowledgments and Disclosure of Funding

KW would like to thank Thomas T.C.K. Zhang and Xinran Zhu for discussions on RKHS. RG is supported by NSF awards OAC-2118201, OAC-1940224 and IIS-1845434. JRG is supported by NSF award IIS-2145644.

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

## A Technical Lemmas

**Lemma 6.** *Let $p_n(x) = \sum_{i=0}^{n} \frac{1}{i!} x^i$ be the degree-$n$ Taylor polynomial of $e^x$ around $x = 0$. Then for any $k \geq 1$ and any $x \in \mathbb{R}$, we have*

$$e^x \geq p_{2k-1}(x).$$

*Proof.* The proof is an induction on $k$. The base case $k = 1$ is trivial: $e^x \geq 1 + x$.

Consider the general case $p_{2k+1}$. Define $g(x) = e^x - p_{2k+1}(x)$. It is easy to see that $g'(x) = e^x - p_{2k}(x)$ and $g''(x) = e^x - p_{2k-1}(x)$. By the induction hypothesis, $g'' \geq 0$ and therefore $g$ is convex. Thus, the minimum of $g$ is given by its stationary points. It is easy to observe that $x = 0$ is indeed a stationary point. Thus, $\min_{x \in \mathbb{R}} g(x) = g(0) = 0$, which finishes the proof. $\square$

**Lemma 7.** *Let $\sigma > 0$ and $W$ be the principal branch of the Lambert $W$ function. For any $m \in \mathbb{N}$, we have*

$$1 + W\left(-\frac{m}{e(m + \sigma^2)}\right) \leq \sqrt{\frac{2\sigma^2}{m + \sigma^2}}.$$

*Proof.* The problem is reduced to proving

$$1 + W(x) \leq \sqrt{2(1 + ex)}$$

for all $-\frac{1}{e} \leq x \leq 0$. In fact, the right hand side is exactly the first-order Taylor expansion of the left hand side at $x = -\frac{1}{e}$ [*e.g.*, 4].

Square both sides. It is equivalent to prove $(1 + W(x))^2 \leq 2(1 + ex)$ for all $x \in [-\frac{1}{e}, 0]$. Define

$$g(x) = (1 + W(x))^2 - 2(1 + ex).$$

The derivative of $g$ in $(-\frac{1}{e}, 0)$ is

$$g'(x) = 2\left(\frac{W(x)}{x} - e\right).$$

By the definition of the Lambert $W$ function, we have

$$ex = W(x)e^{1+W(x)} < W(x)$$

since $-1 < W(x) < 0$ for $x \in (-\frac{1}{e}, 0)$. Thus, $g'(x) < 0$ for $x \in (-\frac{1}{e}, 0)$ and $g$ is a decreasing function in $[-\frac{1}{e}, 0]$. Observe that $g(-\frac{1}{e}) = 0$. Therefore $g(x) \leq 0$ for all $x \in [-\frac{1}{e}, 0]$, which completes the proof. $\square$

Next, we present a lemma which states that the error function bounds the posterior covariance trace in each iteration of Algorithm 1.

**Lemma 8.** *In the $t$-th iteration of Algorithm 1, we have*

$$\text{tr}\left(\nabla k_{\mathcal{D}_t}(\mathbf{x}_t, \mathbf{x}_t)\nabla^\top\right) \leq E_{d,k,\sigma}(b_t)$$

*Proof.* Without loss of generality, we assume $\mathbf{x}_t = \mathbf{0}$. Otherwise, shift the data $\mathcal{D}_t$ and $\mathbf{x}_t$ by $-\mathbf{x}_t$, which does not change the value of the left hand side because of stationarity of the kernel $k$. Let $\mathbf{Z} \in \mathbb{R}^{b_t \times d}$ be arbitrary candidates. Then, we have

$$\text{tr}\left(\nabla k_{\mathcal{D}_{t-1} \cup \mathbf{Z}}(\mathbf{0}, \mathbf{0})\nabla^\top\right) \leq \text{tr}\left(\nabla k(\mathbf{0}, \mathbf{0})\nabla^\top - \nabla k(\mathbf{0}, \mathbf{Z})(k(\mathbf{Z}, \mathbf{Z}) + \sigma^2 \mathbf{I})^{-1} k(\mathbf{Z}, \mathbf{0})\nabla^\top\right).$$

Because the LHS conditions on both $\mathcal{D}_{t-1}$ and $\mathbf{Z}$ but the RHS only conditions on $\mathbf{Z}$. Now, we minimize $\mathbf{Z}$ on both sides.

On the one hand, the LHS becomes $\text{tr}\left(\nabla k_{\mathcal{D}_t}(\mathbf{0}, \mathbf{0})\nabla^\top\right)$. This is because $\mathcal{D}_t$ is the union of $\mathcal{D}_{t-1}$ and the minimizer of the acquisition function $\alpha_{\text{trace}}(\mathbf{0}, \mathbf{Z}) = \text{tr}\left(\nabla k_{\mathcal{D}_{t-1} \cup \mathbf{Z}}(\mathbf{0}, \mathbf{0})\nabla^\top\right)$.

On the other hand, the RHS becomes $E_{d,k,\sigma}(b_t)$ by definition of the error function, which completes the proof.

Note that the edge case, where the minimizer of the acquisition function $\text{argmin}_{\mathbf{Z}} \alpha_{\text{trace}}(\mathbf{0}, \mathbf{Z})$ does not exist (*e.g.*, when $\sigma = 0$), can be handled by a careful limiting argument using the same idea. $\square$

## A.1 Lemmas for the RKHS Assumption

By Assumption 1, the kernel function $k$ is four times continuously differentiable. The following lemma asserts the smoothness of $f \in \mathcal{H}$.

**Lemma 9.** *Suppose $f \in \mathcal{H}$ maps from a compact domain $\mathcal{X}$ to $\mathbb{R}$. Then $f$ is $L$-smooth for some $L$.*

*Proof.* Since the kernel $k$ is four times continuously differentiable, $f$ is twice continuously differentiable. On a compact domain $\mathcal{X}$, the spectral norm of the Hessian $\|\nabla f(\mathbf{x})\|$ has a maximizer. Define $L = \max_{\mathbf{x} \in \mathcal{X}} \|\nabla f(\mathbf{x})\|$. Then $f$ is $L$-smooth. $\square$

**Lemma 1.** *For any $f \in \mathcal{H}$, any $\mathbf{x} \in \mathcal{X}$ and any $\mathcal{D}$, we have the following inequality*

$$\|\nabla f(\mathbf{x}) - \nabla \mu_{\mathcal{D}}(\mathbf{x})\|^2 \leq \mathrm{tr}\big(\nabla k_{\mathcal{D}}(\mathbf{x}, \mathbf{x})\nabla^{\top}\big)\|f\|_{\mathcal{H}}^2. \tag{4}$$

*Proof.* This is a simple corollary of a standard result in meshless scattered data approximation [*e.g.*, 28, Theorem 11.4]. The main idea is to express the estimation error as a linear functional, and then compute the operator norm of that linear functional.

Let $\lambda = \delta_{\mathbf{x}} D : \mathcal{H} \to \mathbb{R}$ be the composition of the evaluation operator and differential operator, *i.e.* $\lambda f = D f(\mathbf{x})$. Wendland [28, Theorem 11.4] provides a bound on $(\lambda f - \lambda \mu_{\mathcal{D}})^2$:

$$(\lambda f - \lambda \mu_{\mathcal{D}})^2 \leq \lambda^{(1)} \lambda^{(2)} k_{\mathcal{D}}(\cdot, \cdot) \|f\|_{\mathcal{H}}^2,$$

where $\lambda^{(1)}$ applies $\lambda$ to the first argument of $k_{\mathcal{D}}$ and $\lambda^{(2)}$ applies $\lambda$ to the second argument of $k_{\mathcal{D}}$.

Pick the linear functional $\lambda : f \mapsto \frac{\partial}{\partial x_i} f(\mathbf{x})$ where $1 \leq i \leq d$. Then, the left hand side becomes $(\lambda f - \lambda \mu_{\mathcal{D}})^2 = (\frac{\partial}{\partial x_i} f(\mathbf{x}) - \frac{\partial}{\partial x_i} \mu_{\mathcal{D}}(\mathbf{x}))^2$. The right hand side $\lambda^{(1)} \lambda^{(2)} k_{\mathcal{D}}(\cdot, \cdot)$ is exactly the $i$-th diagonal entry of $\nabla k_{\mathcal{D}}(\mathbf{x}, \mathbf{x})\nabla^{\top}$. For each $i$, use the above inequality, and then summing over all coordinates finishes the proof. $\square$

## A.2 Lemmas for Convergence on GP Sample Paths

In this section, we provide a few lemmas for the GP sample path $f \sim \mathcal{GP}(0, k)$. By Assumption 1, we have the follow lemma which asserts that $f$ is smooth with high probability.

**Lemma 3.** *For $0 < \delta < 1$, there exists a constant $L > 0$ such that $f$ is $L$-smooth w.p. at least $1 - \delta$.*

*Proof.* The proof uses the Borell-TIS inequality. Let $f \sim \mathcal{GP}(0, k)$ be a Gaussian process. Provided that $\sup_{\mathbf{x} \in \mathcal{X}} |f(\mathbf{x})|$ is almost surely finite, the Borell-TIS inequality states that

$$\Pr(\sup_{\mathbf{x} \in \mathcal{X}} |f(\mathbf{x})| > u + \mathbb{E} \sup_{\mathbf{x} \in \mathcal{X}} |f(\mathbf{x})|) \leq \exp(-\frac{u^2}{2s^2}),$$

where $u > 0$ is an arbitrary positive constant and $s^2 = \sup_{\mathbf{x} \in \mathcal{X}} \mathbb{E}|f(\mathbf{x})|^2$. Namely, if the supremum of $f$ is almost surely finite then the supremum of $f$ is bounded with high probability.

Since $k$ is four time continuously differentiable, the second-order derivative $\frac{\partial^2}{\partial x_i \partial x_j} f$ exists and is almost surely continuous. On a compact domain $\mathcal{X}$, the supremum $\sup_{\mathbf{x} \in \mathcal{X}} |\frac{\partial^2}{\partial x_i \partial x_j} f(\mathbf{x})|$ is almost surely finite. By the Borell-TIS inequality, the expectation $\mathbb{E} \sup_{\mathbf{x} \in \mathcal{X}} |\frac{\partial^2}{\partial x_i \partial x_j} f(\mathbf{x})|$ is finite and the supremum $\sup_{\mathbf{x} \in \mathcal{X}} |\frac{\partial^2}{\partial x_i \partial x_j} f(\mathbf{x})|$ is bounded with high probability. Thus, the Frobenius norm of the Hessian $\nabla^2 f(\mathbf{x})$ is bounded with high probability by a union bound. Since the spectral norm of $\nabla f(\mathbf{x})$ can be bounded by its Frobenius norm, the spectral norm of the Hessian $\|\nabla f(\mathbf{x})\|$ is also bounded with high probability, which gives the smoothness constant. $\square$

The next two lemmas bound the gradient estimation error with high probability.

**Lemma 10.** *Let $\mathbf{u} \sim \mathcal{N}(\mathbf{0}, \boldsymbol{\Sigma})$ be a Gaussian vector. Then for any $t > 0$*

$$\Pr(\|\mathbf{u}\| > t) \leq 2 \exp\left(-\frac{t^2}{2 \mathrm{tr}\, \boldsymbol{\Sigma}}\right).$$

*Proof.* This is a standard concentration inequality result, but we give a self-contained proof here for completeness. Denote the spectral decomposition $\boldsymbol{\Sigma} = \mathbf{Q}\boldsymbol{\Lambda}\mathbf{Q}^\top$. By Markov's inequality, we have

$$\Pr(\|\mathbf{u}\| > t) = \Pr(e^{s\|\mathbf{u}\|} > e^{st})$$
$$\leq e^{-st}\mathbb{E}e^{s\|\mathbf{u}\|},$$

where $s > 0$ is an arbitrary positive constant. Let $\boldsymbol{\epsilon} \sim \mathcal{N}(\mathbf{0}, \mathbf{I})$ be a standard Gaussian variable. Then it is easy to see that $\|\mathbf{u}\| = \|\boldsymbol{\Lambda}^{\frac{1}{2}}\boldsymbol{\epsilon}\|$. Then, replacing $\|\mathbf{u}\|$ with $\|\boldsymbol{\Lambda}^{\frac{1}{2}}\boldsymbol{\epsilon}\|$ gives

$$\Pr(\|\mathbf{u}\| > t) \leq e^{-st}\mathbb{E}e^{s\|\boldsymbol{\Lambda}^{\frac{1}{2}}\boldsymbol{\epsilon}\|}$$
$$\leq e^{-st}\mathbb{E}e^{s\|\boldsymbol{\Lambda}^{\frac{1}{2}}\boldsymbol{\epsilon}\|_1}$$
$$= e^{-st}\prod_{i=1}^d \mathbb{E}e^{s\sqrt{\lambda_i}|\epsilon_i|}$$
$$\leq 2e^{-st}\prod_{i=1}^d \mathbb{E}e^{s\sqrt{\lambda_i}\epsilon_i}$$
$$= 2e^{-st+\frac{1}{2}s^2\sum_{i=1}^d \lambda_i},$$

where the first line plugs in $\boldsymbol{\epsilon}$; the second line uses $\|\cdot\|_2 \leq \|\cdot\|_1$; the third line is due to independence of $\epsilon_i$; the forth line removes the absolute value resulting an extra factor of 2; the last lines uses the moment generating function of $\epsilon_i$. Optimizing the bound over $s$ gives the desired result

$$\Pr(\|\mathbf{u}\| > t) \leq 2e^{-\frac{t^2}{2\sum_{i=1}^d \lambda_i}} = 2\exp(-\frac{t^2}{2\operatorname{tr}\boldsymbol{\Sigma}}).$$

$\square$

**Lemma 11.** *For any $0 < \delta < 1$, let $C_t = 2\log(\frac{\pi^2 t^2}{6\delta})$. Then, the inequalities*

$$\|\nabla f(\mathbf{x}_t) - \nabla\mu_{\mathcal{D}_t}(\mathbf{x}_t)\|^2 \leq C_t \operatorname{tr}\nabla k_{\mathcal{D}_t}(\mathbf{x}_t, \mathbf{x}_t)\nabla^\top$$

*hold for any $t \geq 1$ with probability at least $1 - \delta$.*

*Proof.* Since $\nabla f(\mathbf{x}_t) \sim \mathcal{N}(\nabla\mu_{\mathcal{D}_t}(\mathbf{x}_t), \nabla k_{\mathcal{D}_t}(\mathbf{x}_t, \mathbf{x}_t)\nabla^\top)$, applying Lemma 10 gives

$$\Pr(\|\nabla f(\mathbf{x}_t) - \nabla\mu_{\mathcal{D}_t}(\mathbf{x}_t)\|^2 \geq C_t \operatorname{tr}(\nabla k_{\mathcal{D}_t}(\mathbf{x}_t, \mathbf{x}_t)\nabla^\top)) \leq 2\exp(-\frac{1}{2}C_t).$$

The particular choice of $C_t$ makes the probability on the right hand side become $\frac{6\delta}{\pi^2 t^2}$. Using the union bound over all $t \geq 1$ and using the infinite sum $\sum_{t=1}^\infty \frac{1}{t^2} = \frac{\pi^2}{6}$ finishes the proof. $\square$

We provide an important remark. The probability in Lemma 11 is taken over the randomness of $f$ and the observation noise. On the other hand, the posterior mean gradient $\nabla\mu_{\mathcal{D}_t}$ is deterministic, since it is conditioned on the data $\mathcal{D}_t$.

## B   Bounds on the Error Function $E_{d,k,\sigma}$

This section is devoted to bounding the error function $E_{d,k,\sigma}(b)$ in terms of the batch size $b$. The results in this section immediately give a bound on the posterior covariance trace by Lemma 8.

Before diving into the proofs, we present some immediate corollaries of Assumption 1 on the kernel. Because $k$ is stationary, the kernel can be written as $k(\mathbf{x}, \mathbf{x}') = \phi(\mathbf{x} - \mathbf{x}')$ for some positive-definite function $\phi$. Observe that $\nabla k(\mathbf{x}, \mathbf{x}') = \nabla\phi(\mathbf{x} - \mathbf{x}')$ and $\nabla k(\mathbf{x}, \mathbf{x}')\nabla^\top = -\nabla^2\phi(\mathbf{x} - \mathbf{x}')$. Denote the first-order partial derivative $\partial_i\phi(\mathbf{x}) = \frac{\partial}{\partial x_i}\phi(\mathbf{x})$ and the second-order partial derivative $\partial_i^2\phi(\mathbf{x}) = \frac{\partial^2}{\partial x_i^2}\phi(\mathbf{x})$. It is easy to see that $\phi$ is an even function and $\nabla\phi$ is an odd function. In addition, $\mathbf{0}$ is a maximum of $\phi$. Therefore, $\nabla\phi(\mathbf{0}) = \mathbf{0}$ and the Hessian $\nabla^2\phi(\mathbf{0})$ is negative semi-definite.

## B.1 Noiseless Setting

The following is a bound for the error function $E_{d,k,0}$ for *arbitrary* kernels satisfying Assumption 1 in the noiseless setting $\sigma = 0$.

**Lemma 2.** *For $\sigma = 0$, the error function is bounded by $E_{d,k,0}(b) \leq C \max\{0, 1 + d - b\}$, where $C = \max_{1 \leq i \leq d} \frac{\partial^2}{\partial x_i \partial x_i'} k(\mathbf{0}, \mathbf{0})$ is the maximum of the Hessian's diagonal entries at the origin.*

*Proof.* The bound holds trivially for $b = 0, 1$ and thus a proof is only needed for $b \geq 2$, which we split into two cases $2 \leq b \leq d + 1$ and $b > d + 1$.

We first focus on the case $2 \leq b \leq d + 1$. Let $\mathbf{z}_0 = \mathbf{0}$ and $\mathbf{z}_i = h\mathbf{e}_i$ where $i = 1, 2, \cdots b - 1$, where $\mathbf{e}_i$ is the $i$-th standard unit vector and $h > 0$ is a constant. Define $\mathbf{Z} = (\mathbf{z}_0 \quad \mathbf{z}_1 \quad \cdots \quad \mathbf{z}_{b-1})^\top$. By the definition of the error function, we have

$$E_{d,k,0}(b) \leq \text{tr}\big(\nabla k(\mathbf{0}, \mathbf{0})\nabla^\top - \nabla k(\mathbf{0}, \mathbf{Z})k(\mathbf{Z}, \mathbf{Z})^{-1}k(\mathbf{Z}, \mathbf{0})\nabla^\top\big)$$

$$= \sum_{i=1}^{d} A_{ii}$$

$$= C(1 + d - b) + \sum_{i=1}^{b-1} A_{ii},$$

where we define $\mathbf{A} = \nabla k(\mathbf{0}, \mathbf{0})\nabla^\top - \nabla k(\mathbf{0}, \mathbf{Z})k(\mathbf{Z}, \mathbf{Z})^{-1}k(\mathbf{Z}, \mathbf{0})\nabla^\top$ and use the inequality $A_{ii} \leq -\partial_i^2 \phi(\mathbf{0}) \leq C$ for $b \leq i \leq d$.

Let us focus on the $i$-th diagonal entry $A_{ii}$ where $1 \leq i \leq b - 1$. Then we have

$$A_{ii} \leq -\partial_i^2\phi(\mathbf{0}) - (0 \quad \partial_i\phi(-h\mathbf{e}_i))\begin{pmatrix} \phi(\mathbf{0}) & \phi(h\mathbf{e}_i) \\ \phi(h\mathbf{e}_i) & \phi(\mathbf{0}) \end{pmatrix}^{-1}\begin{pmatrix} 0 \\ \partial_i\phi(-h\mathbf{e}_i) \end{pmatrix}$$

$$= -\partial_i^2\phi(\mathbf{0}) - \frac{1}{(\phi(\mathbf{0}))^2 - (\phi(h\mathbf{e}_i))^2}(0 \quad \partial_i\phi(-h\mathbf{e}_i))\begin{pmatrix} \phi(\mathbf{0}) & -\phi(h\mathbf{e}_i) \\ -\phi(h\mathbf{e}_i) & \phi(\mathbf{0}) \end{pmatrix}\begin{pmatrix} 0 \\ \partial_i\phi(-h\mathbf{e}_i) \end{pmatrix}$$

$$= -\partial_i^2\phi(\mathbf{0}) - \frac{\phi(\mathbf{0})(\partial_i\phi(h\mathbf{e}_i))^2}{(\phi(\mathbf{0}))^2 - (\phi(h\mathbf{e}_i))^2},$$

where the first line is because conditioning on the subset $\mathbf{z}_0$ and $\mathbf{z}_i$ does not make the posterior smaller. Now let $h \to 0$ and compute the limit by L'Hôpital's rule. We have

$$\lim_{h \to 0} A_{ii}(h) = \lim_{h \to 0} -\partial_i^2\phi(\mathbf{0}) - \frac{\phi(\mathbf{0})(\partial_i\phi(h\mathbf{e}_i))^2}{(\phi(\mathbf{0}))^2 - (\phi(h\mathbf{e}_i))^2}$$

$$= \lim_{h \to 0} -\partial_i^2\phi(\mathbf{0}) - \frac{\phi(\mathbf{0})}{\phi(\mathbf{0}) + \phi(h\mathbf{e}_i)} \cdot \frac{2\partial_i\phi(h\mathbf{e}_i)\partial_i^2\phi(h\mathbf{e}_i)}{-\partial_i\phi(h\mathbf{e}_i)}$$

$$= 0.$$

Thus letting $h \to 0$ gives the inequality $E_{d,k,\sigma}(b) \leq C(1 + d - b)$ for $2 \leq b \leq d + 1$.

When $d > d + 1$, note that $E_{d,k,\sigma}(b)$ is an decreasing function in $b$ and thus $E_{d,k,\sigma}(b) \leq E_{d,k,\sigma}(d + 1) = 0$. Both cases can be bounded by the expression $C \max\{0, 1 + d - b\}$. $\qquad\square$

## B.2 Noisy Setting

This section proves bounds on the error function $E_{d,k,\sigma}$ for the RBF kernel and the $\nu = \frac{5}{2}$ Matérn kernel in the noisy setting. The lemmas in this section will implicitly use the assumption that $k(\mathbf{0}, \mathbf{0}) = 1$. This assumption is indeed satisfied by the RBF kernel and the Matérn kernel, which are of primary concern in this paper.

Before proving the bound on $E_{d,k,\sigma}$, we need one more technical lemma:

**Lemma 12** (Central Differencing Designs). *Consider the $2md$ points $\mathbf{Z} \in \mathbb{R}^{2md \times d}$ defined as*

$$\mathbf{z}_j^{(i)} = \begin{cases} -h\mathbf{e}_i, & j = 1, 2, \cdots m \\ h\mathbf{e}_i, & j = m + 1, m + 2 \cdots 2m, \end{cases}$$

*where $1 \leq i \leq d$, $1 \leq j \leq 2m$ and $\mathbf{e}_i$ is the $i$-th standard unit vector. Define*

$$\mathbf{A} = \nabla k(\mathbf{0}, \mathbf{0})\nabla^\top - \nabla k(\mathbf{0}, \mathbf{Z})(k(\mathbf{Z}, \mathbf{Z}) + \sigma^2\mathbf{I})^{-1}k(\mathbf{Z}, \mathbf{0})\nabla^\top.$$

*Then, we have*

$$A_{ii} \leq -\partial_i^2\phi(\mathbf{0}) - \frac{2\beta_i^2}{(1-\alpha_i)+\gamma},$$

*for all $1 \leq i \leq d$ and thus*

$$\mathrm{tr}(\mathbf{A}) \leq -\sum_{i=1}^d (\partial_i^2\phi(\mathbf{0}) + \frac{2\beta_i^2}{(1-\alpha_i)+\gamma}),$$

*where $\alpha_i = \phi(2h\mathbf{e}_i)$, $\beta_i = \partial_i\phi(-h\mathbf{e}_i)$ and $\gamma = \frac{\sigma^2}{m}$.*

*Proof.* Note that $\mathbf{A}$ is the posterior covariance at the origin $\mathbf{0}$ conditioned on $\mathbf{Z}$. Denote $\mathbf{Z}^{(i)} = \begin{pmatrix} \mathbf{z}_1^{(i)} & \mathbf{z}_2^{(i)} & \cdots & \mathbf{z}_m^{(i)} & \mathbf{z}_{m+1}^{(i)} & \cdots & \mathbf{z}_{2m}^{(i)} \end{pmatrix}^\top$ the subset of $2m$ points that lie on the $i$-th axis. Then, the $i$-th diagonal entry $A_{ii}$ can be bounded by

$$A_{ii} \leq -\partial_i^2\phi(\mathbf{0}) - \partial_i k(\mathbf{0}, \mathbf{Z}^{(i)})(k(\mathbf{Z}^{(i)}, \mathbf{Z}^{(i)}) + \sigma^2\mathbf{I})^{-1}(\partial_i k(\mathbf{0}, \mathbf{Z}^{(i)}))^\top,$$

since conditioning on only a subset of points $\mathbf{Z}^{(i)}$ would not make the posterior variances smaller (*e.g.*, the posterior covariance is the Schur complement of a positive definite matrix). The remaining proof is dedicated to bounding the right hand side.

First, we need to compute the inverse of the $2m \times 2m$ kernel matrix

$$\widehat{\mathbf{K}} = k(\mathbf{Z}^{(i)}, \mathbf{Z}^{(i)}) + \sigma^2\mathbf{I}$$
$$= \begin{pmatrix} \mathbf{1}\mathbf{1}^\top & \alpha_i\mathbf{1}\mathbf{1}^\top \\ \alpha_i\mathbf{1}\mathbf{1}^\top & \mathbf{1}\mathbf{1}^\top \end{pmatrix} + \sigma^2\mathbf{I},$$

where $\alpha_i = \phi(2h\mathbf{e}_i)$ is a nonnegative constant and $\mathbf{1}$ is a $m$ dimensional vector (we drop the index $i$ in the matrix $\widehat{\mathbf{K}}$ for notation simplicity). We compute the inverse analytically by forming its eigendecomposition

$$\widehat{\mathbf{K}} = \mathbf{Q}\mathbf{\Lambda}\mathbf{Q}^\top,$$

where $\mathbf{\Lambda} = \mathrm{diag}(\lambda_1, \lambda_2, \cdots, \lambda_{2m})$ and $\mathbf{Q} = \begin{pmatrix} \mathbf{q}_1 & \mathbf{q}_2 & \cdots & \mathbf{q}_{2m} \end{pmatrix}$. Observe that:

$$\widehat{\mathbf{K}} = \begin{pmatrix} 1 & \alpha_i \\ \alpha_i & 1 \end{pmatrix} \otimes \mathbf{1}\mathbf{1}^\top + \sigma^2\mathbf{I},$$

where $\otimes$ denotes the Kronecker product. Because the eigenvalues (vectors) of a Kronecker product equal the Kronecker product of the individual eigenvalues (vectors), and because adding a diagonal shift simply shifts the eigenvalues, the top two eigenvalues of $\widehat{\mathbf{K}}$ are $\lambda_1 = m(1 + \alpha_i) + \sigma^2$ and $\lambda_2 = m(1 - \alpha_i) + \sigma^2$. The remaining $2m - 2$ eigenvalues are $\sigma^2$. The top two eigenvectors are

$$\mathbf{q}_1 = \frac{1}{\sqrt{2m}}\begin{pmatrix} \mathbf{1} \\ \mathbf{1} \end{pmatrix}, \quad \mathbf{q}_2 = \frac{1}{\sqrt{2m}}\begin{pmatrix} \mathbf{1} \\ -\mathbf{1} \end{pmatrix},$$

Next, we cope with the term $\partial_i k(\mathbf{0}, \mathbf{Z}^{(i)})$, where the partial derivative is taken w.r.t. the first argument's $i$-th coordinate. Denote $\mathbf{v}^\top = \partial_i k(\mathbf{0}, \mathbf{Z}^{(i)})$. Then it is easy to see that

$$\mathbf{v} = \beta_i\begin{pmatrix} -\mathbf{1} \\ \mathbf{1} \end{pmatrix},$$

where $\beta_i = \partial_i\phi(-h\mathbf{e}_i)$. Note that $\mathbf{v}$ happens to be an eigenvector of $\widehat{\mathbf{K}}$ as well, because $\mathbf{v} \parallel \mathbf{q}_2$. As a result, $\mathbf{Q}^\top\mathbf{v}$ has a simple expression $\mathbf{Q}^\top\mathbf{v} = \begin{pmatrix} 0 & -\sqrt{2m}\beta_i & 0 & \cdots & 0 \end{pmatrix}^\top$. Thus, we have

$$A_{ii} \leq -\partial_i^2\phi(\mathbf{0}) - \mathbf{v}^\top\mathbf{Q}\mathbf{\Lambda}^{-1}\mathbf{Q}^\top\mathbf{v}$$
$$= -\partial_i^2\phi(\mathbf{0}) - \frac{2m\beta_i^2}{m(1-\alpha_i)+\sigma^2}$$
$$= -\partial_i^2\phi(\mathbf{0}) - \frac{2\beta_i^2}{(1-\alpha_i)+\gamma}.$$

Summing over the coordinates $1 \leq i \leq d$ finishes the proof. $\qquad\square$

With Lemma 12, we are finally ready to present the bounds for the RBF kernel and the Matérn kernel.

**Lemma 4** (RBF Kernel). *Let $k(\mathbf{x}_1, \mathbf{x}_2) = \exp\left(-\frac{1}{2}\|\mathbf{x}_1 - \mathbf{x}_2\|^2\right)$ be the RBF kernel. We have*

$$E_{d,k,\sigma}(2md) \leq d\left(1 + W\left(-\frac{m}{e(m+\sigma^2)}\right)\right) = \mathcal{O}(\sigma d m^{-\frac{1}{2}}),$$

*where $m \in \mathbb{N}$ and $W$ denotes the principal branch of the Lambert W function.*

*Proof.* For any $\mathbf{Z} \in \mathbb{R}^{2md \times d}$, the following inequality holds by the definition of error function

$$E_{d,k,\sigma}(b) \leq \operatorname{tr}\left(\nabla k(\mathbf{0}, \mathbf{0})\nabla^\top - \nabla k(\mathbf{0}, \mathbf{Z})(k(\mathbf{Z}, \mathbf{Z}) + \sigma^2\mathbf{I})^{-1}k(\mathbf{Z}, \mathbf{0})\nabla^\top\right).$$

Consider the following points $\mathbf{z}_j^{(i)}$ defined as

$$\mathbf{z}_j^{(i)} = \begin{cases} -h\mathbf{e}_i, & j = 1, 2, \cdots m \\ h\mathbf{e}_i, & j = m+1, m+2 \cdots 2m, \end{cases}$$

where $1 \leq i \leq d$. The total number of points is exactly $2md$. By Lemma 12, we have

$$E_{d,k,\sigma}(b) \leq -\sum_{i=1}^d \left(\partial_i^2\phi(\mathbf{0}) + \frac{2\beta_i^2}{(1-\alpha_i) + \gamma}\right).$$

For the RBF kernel, the values of $\alpha_i$ and $\beta_i$ are the same for each coordinate $1 \leq i \leq d$ since it is isotropic:

$$\alpha = \alpha_i = \phi(2h\mathbf{e}_i) = \exp(-2h^2), \quad \beta = \beta_i = \partial_i\phi(-h\mathbf{e}_i) = \exp\left(-\frac{1}{2}h^2\right)h.$$

Plugging the value of $\alpha$, $\beta$, $\gamma$ and $\partial_i^2\phi(\mathbf{0})$ into the bound on $E_{d,k,\sigma}$, we have

$$A_{ii} \leq 1 - \frac{2m\exp(-h^2)h^2}{m(1 - \exp(-2h^2)) + \sigma^2}$$
$$\leq 1 - \frac{2m\exp(-2h^2)h^2}{m(1 - \exp(-2h^2)) + \sigma^2}$$

where the second inequality replaces $\exp(-h^2)$ with $\exp(-2h^2)$ in the numerator. Because the bound holds for arbitrary $h$, we can apply the transformation $h \mapsto \frac{1}{\sqrt{2}}h$, which gives the inequality

$$A_{ii} \leq 1 - \frac{m\exp(-h^2)h^2}{m(1 - \exp(-h^2)) + \sigma^2}.$$

Our goal is to bound $A_{ii}$ in terms of $m$ and $\sigma^2$. Therefore, we minimize the right hand side over $h$. Define $g(x) = 1 - \frac{me^{-x}x}{m(1-e^{-x})+\sigma^2}$, where $x \geq 0$. The derivative is given by:

$$g'(x) = \frac{m(m + (m+\sigma^2)e^x(x-1))}{(m(e^x - 1) + \sigma^2 e^x)^2}.$$

The unique stationary point is $x^* = 1 + W\left(-\frac{m}{e(m+\sigma^2)}\right)$, where $W$ is the principal branch of the Lambert $W$ function. It is easy to see the stationary point $x^*$ is the global minimizer of $g(x)$ over $\mathbb{R}_{\geq 0}$. Plug $x^*$ into the expression of $g$. Coincidentally, we have $g(x^*) = 1 + W\left(-\frac{m}{e(m+\sigma^2)}\right)$ as well — $x^*$ is a fixed point of $g$.

In summary, we have shown $A_{ii} \leq 1 + W\left(-\frac{m}{e(m+\sigma^2)}\right)$ for each coordinate $i$. Summing $A_{ii}$ over all $d$ coordinates proves the first inequality $E_{d,k,\sigma}(2md) \leq d\left(1 + W\left(-\frac{m}{e(m+\sigma^2)}\right)\right)$. The second inequality is a direction implication of Lemma 7, which completes the proof. $\square$

**Lemma 5** (Matern Kernel). *Let $k(\cdot, \cdot)$ be the $\nu = 2.5$ Matérn kernel. Then, we have*

$$E_{d,k,\sigma}(2md) \lesssim \sigma d m^{-\frac{1}{2}} + \sigma^{\frac{3}{2}} d m^{-\frac{3}{4}} = \mathcal{O}(\sigma d m^{-\frac{1}{2}}).$$

*Proof.* The proof is similar to Lemma 4. The difference is that we need to upper bound $\partial_i^2\phi(\mathbf{0}) + \frac{2\beta_i^2}{(1-\alpha_i)+\gamma}$ by a rational function. Otherwise the expression is intractable to minimize.

The Matérn kernel with half integer $\nu$ can be written as a product of an exponential and a polynomial. In particular, for $\nu = \frac{5}{2}$, we have

$$\phi(\mathbf{x}) = (1 + \sqrt{5}\|\mathbf{x}\| + \frac{5}{3}\|\mathbf{x}\|^2)\exp(-\sqrt{5}\|\mathbf{x}\|).$$

When $\mathbf{x}$ is in the nonnegative orthant, the gradient is

$$\nabla\phi(\mathbf{x}) = \exp(-\sqrt{5}\|\mathbf{x}\|)(\frac{\sqrt{5}}{\|\mathbf{x}\|}\mathbf{x} + \frac{10}{3}\mathbf{x}) - \frac{\sqrt{5}}{\|\mathbf{x}\|}\exp(-\sqrt{5}\|\mathbf{x}\|)(1 + \sqrt{5}\|\mathbf{x}\| + \frac{5}{3}\|\mathbf{x}\|^2)\mathbf{x}$$

$$= -\frac{5}{3}\exp(-\sqrt{5}\|\mathbf{x}\|)(1 + \sqrt{5}\|\mathbf{x}\|)\mathbf{x}.$$

Since the Matérn kernel is isotropic, the $\alpha_i$ and $\beta_i$ as in Lemma 12 are the same across different coordinate $i$, and their values are

$$\alpha = \alpha_i = \phi(2h\mathbf{e}_i) = \exp(-2\sqrt{5}h)(1 + 2\sqrt{5}h + \frac{20}{3}h^2),$$

$$\beta = \beta_i = \partial_i\phi(-h\mathbf{e}_i) = -\partial_i\phi(h\mathbf{e}_i) = \frac{5}{3}\exp(-\sqrt{5}h)(1 + \sqrt{5}h)h,$$

In addition, $-\partial_i^2\phi(\mathbf{0}) = \frac{5}{3}$. By Lemma 12, we have

$$A_{ii} \le -\partial_i^2\phi(\mathbf{0}) - \frac{2\beta^2}{(1-\alpha)+\gamma}$$

$$= \frac{5}{3} - \frac{10\exp(-2h)(1+h)^2h^2}{3(3 - \exp(-2h)(3 + 6h + 4h^2)) + 9\gamma}.$$

Next, we approximate the exponential function $\exp(-2h)$ by its Taylor polynomials. By Lemma 6, use the inequality $\exp(-2h) \ge 1 - 2h$ for the numerator and the inequality $\exp(-2h) \ge 1 - 2h + 2h^2 - \frac{4}{3}h^3$ for the denominator. Applying these two inequalities gives

$$A_{ii} \le \frac{5}{3} - \frac{10(1-2h)(1+h)^2h^2}{2h^2(3 + 8h^3) + 9\gamma}.$$

Let $h = \gamma^{\frac{1}{4}}$ and thus $\gamma = h^4$. Then we have

$$A_{ii} \le \frac{5}{3} - \frac{10(1-2h)(1+h)^2h^2}{2h^2(3 + 8h^3) + 9h^4}$$

$$= \frac{5h^2(27 + 28h)}{3(6 + 9h^2 + 16h^3)}$$

$$\le \frac{5}{18}h^2(27 + 28h)$$

$$= \frac{15}{2}\gamma^{\frac{1}{2}} + \frac{70}{9}\gamma^{\frac{3}{4}}$$

$$= \frac{15}{2}\sigma m^{-\frac{1}{2}} + \frac{70}{9}\sigma^{\frac{3}{2}}m^{-\frac{3}{4}}$$

where the third line drops $h^2$ and $h^3$ in the denominator; the forth line plugs in the value $h = \gamma^{\frac{1}{4}}$ back and drops the constants. Summing over the coordinates $1 \le i \le d$ gives the first inequality:

$$E_{d,k,\sigma}(2md) \lesssim \sigma dm^{-\frac{1}{2}} + \sigma^{\frac{3}{2}}dm^{-\frac{3}{4}}$$

For large enough $m$, the bound is dominated by the first term $\sigma dm^{-\frac{1}{2}}$, which completes the proof. $\square$

We end this section with a short summary. Lemma 4 and Lemma 5 happen to end up with the same rate $E_{d,k,s}(2md) = \mathcal{O}(\sigma dm^{-\frac{1}{2}})$. Replacing $2md$ with the batch size $b$, we have shown that $E_{d,k,s}(b) = \mathcal{O}(\sigma d^{\frac{3}{2}}b^{-\frac{1}{2}})$ for the RBF kernel and $\nu = 2.5$ Matérn kernel.

### B.3 Discussion: Forward Differencing Designs

This section explores an alternative proof for the error function based on forward differencing designs, as opposed to the central differencing designs in Lemma 12. Similar to the previous section, we assume $k(\mathbf{0}, \mathbf{0}) = 1$, which is indeed satisfied by the RBF kernel and Matérn kernel.

**Lemma 13.** *Consider following $(d+1)m$ points $\mathbf{Z} \in \mathbb{R}^{(d+1)m \times d}$ defined as*

$$
\begin{aligned}
\mathbf{z}_j^{(0)} &= \mathbf{0}, & j &= 1, 2, \cdots, m \\
\mathbf{z}_j^{(i)} &= h\mathbf{e}_i, & i \geq 1, & \quad j = 1, 2, \cdots, m
\end{aligned}
$$

*where $\mathbf{e}_i$ is the $i$-th standard unit vector. Define*

$$
\mathbf{A} = \nabla k(\mathbf{0}, \mathbf{0})\nabla^\top - \nabla k(\mathbf{0}, \mathbf{Z})(k(\mathbf{Z}, \mathbf{Z}) + \sigma^2\mathbf{I})^{-1}k(\mathbf{Z}, \mathbf{0})\nabla^\top.
$$

*Then, we have*

$$
\operatorname{tr}\mathbf{A} \leq -\sum_{i=1}^{d}\left(\partial_i^2\phi(\mathbf{0}) + \frac{(1+\gamma)\beta_i^2}{(1+\gamma)^2 - \alpha_i^2}\right)
$$

*where $\alpha_i = \phi(h\mathbf{e}_i)$, $\beta_i = \partial_i\phi(-h\mathbf{e}_i)$ and $\gamma = \frac{\sigma^2}{m}$.*

*Proof.* The proof is similar to Lemma 12.

Denote $\mathbf{Z}^{(i)}$ be the subset of points consisting of $\mathbf{z}_j^{(i)}$ and $\mathbf{z}_j^{(0)}$, where $j = 1, 2, \cdots, m$. Notice that the $i$-th diagonal entry $A_{ii}$ can be bounded by

$$
A_{ii} \leq -\partial_i^2\phi(\mathbf{0}) - \partial_i k(\mathbf{0}, \mathbf{Z}^{(i)})(k(\mathbf{Z}^{(i)}, \mathbf{Z}^{(i)}) + \sigma^2\mathbf{I})^{-1}(\partial_i k(\mathbf{0}, \mathbf{Z}^{(i)}))^\top.
$$

We need to invert the $2m \times 2m$ matrix

$$
\begin{aligned}
\widehat{\mathbf{K}} &= k(\mathbf{Z}^{(i)}, \mathbf{Z}^{(i)}) + \sigma^2\mathbf{I} \\
&= \begin{pmatrix} \mathbf{1}\mathbf{1}^\top & \alpha_i\mathbf{1}^\top \\ \alpha_i\mathbf{1} & \mathbf{1}\mathbf{1}^\top \end{pmatrix} + \sigma^2\mathbf{I}.
\end{aligned}
$$

Again, we resort to the eigendecomposition of $\widehat{\mathbf{K}} = \mathbf{Q}\mathbf{\Lambda}\mathbf{Q}^\top$. The top two eigenvalues of $\widehat{\mathbf{K}}$ are $\lambda_1 = m(1 + \alpha_i) + \sigma^2$ and $\lambda_2 = m(1 - \alpha_i) + \sigma^2$. The remaining $2m - 2$ eigenvalues are $\sigma^2$. The top two eigenvectors are

$$
\mathbf{q}_1 = \frac{1}{\sqrt{2m}}\begin{pmatrix} \mathbf{1} \\ \mathbf{1} \end{pmatrix}, \quad \mathbf{q}_2 = \frac{1}{\sqrt{2m}}\begin{pmatrix} \mathbf{1} \\ -\mathbf{1} \end{pmatrix}.
$$

Denote $\mathbf{v}^\top = \partial_i k(\mathbf{0}, \mathbf{Z}^{(i)})$. Note that $\mathbf{v}$ can be written as a linear combination of $\mathbf{q}_1$ and $\mathbf{q}_2$:

$$
\mathbf{v} = \beta_i\begin{pmatrix} \mathbf{0} \\ \mathbf{1} \end{pmatrix} = \frac{1}{2}\beta_i\sqrt{2m}(\mathbf{q}_1 - \mathbf{q}_2).
$$

Then, straightforward calculation gives $\mathbf{Q}^\top\mathbf{v} = \frac{1}{2}\beta_i\sqrt{2m}(1 \quad -1 \quad 0 \quad \cdots \quad 0)^\top$.

Then, we have

$$
\begin{aligned}
A_{ii} &\leq -\partial_i^2\phi(\mathbf{0}) - \mathbf{v}^\top\mathbf{Q}\mathbf{\Lambda}^{-1}\mathbf{Q}^\top\mathbf{v} \\
&= -\partial_i^2\phi(\mathbf{0}) - \frac{1}{2}m\beta_i^2\left(\frac{1}{m(1+\alpha_i) + \sigma^2} + \frac{1}{m(1-\alpha_i) + \sigma^2}\right) \\
&= -\partial_i^2\phi(\mathbf{0}) - \frac{1}{2}\beta_i^2\left(\frac{1}{1 + \alpha_i + \gamma} + \frac{1}{1 - \alpha_i + \gamma}\right) \\
&= -\partial_i^2\phi(\mathbf{0}) - \frac{\beta_i^2(1+\gamma)}{(1+\gamma)^2 - \alpha_i^2}.
\end{aligned}
$$

Summing over all coordinates finishes the proof. $\qquad\square$

**Lemma 14.** *Let $k(\mathbf{x}_1, \mathbf{x}_2) = \exp\left(-\frac{1}{2}\|\mathbf{x}_1 - \mathbf{x}_2\|^2\right)$ be the RBF kernel. The forward differencing designs give a decay rate of $E_{d,k,\sigma}(b) = \mathcal{O}(\sigma d m^{-\frac{1}{2}})$.*

*Proof.* Define $\alpha = \phi(h\mathbf{e}_i)$ and $\beta = \partial_i \phi(-h\mathbf{e}_i)$. Plugging the values of $\alpha$ and $\beta$ into the bounds in Lemma 13 yields

$$A_{ii} \leq 1 - \frac{h^2 \exp(-h^2)(1+\gamma)}{(1+\gamma)^2 - \exp(-h^2)}.$$

The bound holds for arbitrary $h$. Applying the transformation $h \mapsto \sqrt{h}$ gives

$$
\begin{aligned}
A_{ii} &\leq 1 - \frac{h \exp(-h)(1+\gamma)}{(1+\gamma)^2 - \exp(-h)} \\
&\leq 1 - \frac{h(1-h)(1+\gamma)}{(1+\gamma)^2 - (1-h)} \\
&= 1 - \frac{h(1-h)(1+\gamma)}{\gamma^2 + 2\gamma + h} \\
&\leq 1 - \frac{h(1-h)}{\gamma^2 + 2\gamma + h},
\end{aligned}
$$

where the second line uses the inequality $\exp(-h) \geq 1 - h$ in the numerator and the denominator; the last line is because $\gamma$ is nonnegative. Let $h = \gamma^{\frac{1}{2}}$ so that $\gamma = h^2$. Then we have

$$
\begin{aligned}
A_{ii} &\leq 1 - \frac{h(1-h)}{h^4 + 2h^2 + h} \\
&= \frac{h^3 + 3h}{h^3 + 2h + 1} \\
&\leq h^3 + 3h \\
&= \gamma^{\frac{3}{2}} + 3\gamma^{\frac{1}{2}} \\
&\lesssim \gamma^{\frac{1}{2}} \\
&= \sigma m^{-\frac{1}{2}}
\end{aligned}
$$

where the first line plugs in $\gamma = h^2$; the third line drops the $h^3 + 2h$ in the denominator; the forth lines plugs in $h = \gamma^{\frac{1}{2}}$; the fifth line is because $\gamma^{\frac{1}{2}}$ dominates the bound when $m$ is large. Thus, we have shown $E_{d,k,\sigma}(dm + m) = \mathcal{O}(\sigma d m^{-\frac{1}{2}})$. $\qquad\square$

The above lemma shows that forward differencing designs achieve the same asymptotic decay rate as the central differencing designs for the RBF kernel. Though, the leading constant in the big $\mathcal{O}$ notation is slightly larger.

## C   Convergence Proofs

The following is a useful lemma for biased gradient updates, which bounds the gradient norm via the cumulative bias. The proof is adapted from a lemma by Ajalloeian and Stich [1, Lemma 2].

**Lemma 15.** *Let $f$ be $L$-smooth and bounded from below. Suppose the gradient oracle $\hat{\mathbf{g}}_t$ has bias bounded by $\xi_t$ in the $t$-th iteration. Namely, we have $\|\hat{\mathbf{g}}_t - \mathbf{g}_t\|^2 \leq \xi_t$ for all $t \geq 0$, where $\mathbf{g}_t = \nabla f(\mathbf{x}_t)$ is the ground truth gradient. Then the update $\mathbf{x}_{t+1} = \mathbf{x}_t - \eta_t \hat{\mathbf{g}}_t$ with $\eta_t \leq \frac{1}{L}$ produces a sequence $\{\mathbf{x}_t\}_{t=1}^{\infty}$ satisfying*

$$\min_{1 \leq t \leq T} \|\nabla f(\mathbf{x}_t)\|^2 \leq \frac{2(f(\mathbf{x}_1) - f^*)}{\sum_{t=1}^{T} \eta_t} + \frac{\sum_{i=1}^{T} \eta_t \xi_t}{\sum_{t=1}^{T} \eta_t}$$

*Proof.* By $L$-smoothness, we have

$$f(\mathbf{x}_{t+1}) \leq f(\mathbf{x}_t) + \nabla f(\mathbf{x}_t)^\top (\mathbf{x}_{t+1} - \mathbf{x}_t) + \frac{1}{2}L\|\mathbf{x}_{t+1} - \mathbf{x}_t\|^2.$$

Plugging in the update formula $\mathbf{x}_{t+1} = \mathbf{x}_t - \eta_t \hat{\mathbf{g}}_t$, we have

$$
\begin{aligned}
f(\mathbf{x}_{t+1}) &\leq f(\mathbf{x}_t) - \eta_t \nabla f(\mathbf{x}_t)^\top \hat{\mathbf{g}}_t + \frac{1}{2}L\eta_t^2 \|\hat{\mathbf{g}}_t\|^2 \\
&\leq f(\mathbf{x}_t) - \eta_t \nabla f(\mathbf{x}_t)^\top \hat{\mathbf{g}}_t + \frac{1}{2}\eta_t \|\hat{\mathbf{g}}_t\|^2 \\
&\leq f(\mathbf{x}_t) - \eta_t \nabla f(\mathbf{x}_t)^\top \hat{\mathbf{g}}_t + \frac{1}{2}\eta_t \|\hat{\mathbf{g}}_t - \nabla f(\mathbf{x}_t) + \nabla f(\mathbf{x}_t)\|^2 \\
&\leq f(\mathbf{x}_t) + \frac{1}{2}\eta_t (\|\hat{\mathbf{g}}_t - \nabla f(\mathbf{x}_t)\|^2 - \|\nabla f(\mathbf{x}_t)\|^2) \\
&\leq f(\mathbf{x}_t) - \frac{1}{2}\eta_t \|\nabla f(\mathbf{x}_t)\|^2 + \frac{1}{2}\eta_t \xi_t,
\end{aligned}
$$

where the first inequality uses $L$-smoothness; the second inequality uses $\eta_t \leq \frac{1}{L}$; the fourth inequality expands the squared Euclidean norm; the last inequality uses the definition of bias. Summing the inequalities for $t = 1, 2, \cdots, T$ and rearranging the terms, we have

$$
\begin{aligned}
\sum_{t=1}^{T} \eta_t \|\nabla f(\mathbf{x}_t)\|^2 &\leq 2(f(\mathbf{x}_1) - f(\mathbf{x}_{T+1})) + \sum_{t=1}^{T} \eta_t \xi_t \\
&\leq 2(f(\mathbf{x}_1) - f^*) + \sum_{t=1}^{T} \eta_t \xi_t.
\end{aligned}
$$

Dividing both sides by $\sum_{t=1}^{T} \eta_t$ gives

$$\frac{\sum_{t=1}^{T} \eta_t \|\nabla f(\mathbf{x}_t)\|^2}{\sum_{t=1}^{T} \eta_t} \leq \frac{2(f(\mathbf{x}_1) - f^*)}{\sum_{t=1}^{T} \eta_t} + \frac{\sum_{t=1}^{T} \eta_t \xi_t}{\sum_{t=1}^{T} \eta_t}.$$

The left hand side is a weighted average, which is greater than the minimum over $1 \leq t \leq T$, which completes the proof. $\qquad \square$

Next, we prove a variant of Lemma 15 for the projected update $\mathbf{x}_{t+1} = \mathrm{proj}_{\mathcal{X}}(\mathbf{x}_t - \eta_t \nabla f(\mathbf{x}_t))$. For the ground truth gradient $\mathbf{g}_t = \nabla f(\mathbf{x}_t)$, define the gradient mapping

$$G(\mathbf{x}_t) = \frac{1}{\eta_t}\big(\mathbf{x}_t - \mathrm{proj}_{\mathcal{X}}(\mathbf{x}_t - \eta_t \mathbf{g}_t)\big).$$

For the approximate gradient $\hat{\mathbf{g}}_t = \nabla \mu_{\mathcal{D}_t}(\mathbf{x}_t)$, define the gradient mapping

$$\widehat{G}(\mathbf{x}_t) = \frac{1}{\eta_t}\big(\mathbf{x}_t - \mathrm{proj}_{\mathcal{X}}(\mathbf{x}_t - \eta_t \hat{\mathbf{g}}_t)\big).$$

In the following, we introduce two lemmas characterizing the projection operator $\mathrm{proj}_{\mathcal{X}}(\cdot)$.

**Lemma 16** (*e.g.*, Lemma 3.1 of Bubeck et al. [2])**.** *Let $\mathcal{X}$ be convex and compact. Let $\mathbf{x} \in \mathcal{X}$ and $\mathbf{z} \in \mathbb{R}^d$. Then we have*

$$(\mathrm{proj}_{\mathcal{X}}(\mathbf{z}) - \mathbf{z})^\top (\mathbf{x} - \mathrm{proj}_{\mathcal{X}}(\mathbf{z})) \geq 0.$$

*As a result, $\|\mathbf{x} - \mathbf{z}\|^2 \geq \|\mathrm{proj}_{\mathcal{X}}(\mathbf{z}) - \mathbf{z}\|^2 + \|\mathbf{x} - \mathrm{proj}_{\mathcal{X}}(\mathbf{z})\|^2.$*

**Lemma 17.** *The following holds:*

1. $\|G(\mathbf{x}_t)\| \leq \|\mathbf{g}_t\|$,

2. $\|G(\mathbf{x}_t) - \mathbf{g}\| \leq \|\mathbf{g}_t\|$,

3. $\|\widehat{G}(\mathbf{x}_t) - G(\mathbf{x}_t)\| \leq \|\hat{\mathbf{g}}_t - \mathbf{g}_t\|$,

*4.* $\mathbf{g}^\top G(\mathbf{x}_t) \geq \|G(\mathbf{x}_t)\|^2.$

*Proof.* (1-2): The first two inequalities are direct corollaries of Lemma 16.

(3): This is proved as follows:

$$\|\widehat{G}(\mathbf{x}_t) - G(\mathbf{x}_t)\| = \frac{1}{\eta_t}\|\mathrm{proj}_{\mathcal{X}}(\mathbf{x}_t - \eta_t\mathbf{g}_t) - \mathrm{proj}_{\mathcal{X}}(\mathbf{x}_t - \eta_t\hat{\mathbf{g}}_t)\|$$
$$\leq \|\hat{\mathbf{g}}_t - \mathbf{g}_t\|,$$

where the second line is because the projection operator is non-expansive.

(4): By Lemma 16, we have

$$-\eta_t G(\mathbf{x}_t)^\top(\eta_t G(\mathbf{x}_t) - \eta_t\mathbf{g}_t) \geq 0.$$

Rearranging the terms finishes the proof. $\qquad\square$

Now we give a lemma proving biased gradient update with the projection operator. The proof is adapted from a lemma by Shu et al. [22].

**Lemma 18.** *Let $f$ be $L$-smooth over a convex compact set $\mathcal{X}$. Moreover, assume the gradient norm $\|\nabla f(\mathbf{x})\|$ is bounded by $L'$ on $\mathcal{X}$. Suppose the gradient oracle $\hat{\mathbf{g}}_t$ has bias bounded by $\xi_t$ in the $t$-th iteration: $\|\hat{\mathbf{g}}_t - \mathbf{g}_t\|^2 \leq \xi_t$ for all $t \geq 0$, where $\mathbf{g}_t = \nabla f(\mathbf{x}_t)$. Then the update $\mathbf{x}_{t+1} = \mathrm{proj}_{\mathcal{X}}(\mathbf{x}_t - \eta_t\hat{\mathbf{g}}_t)$ with $\eta_t \leq \frac{1}{L}$ produces a sequence $\{\mathbf{x}_t\}_{t=1}^\infty$ satisfying*

$$\min_{1 \leq t \leq T}\|G(\mathbf{x}_t)\|^2 \leq \frac{2(f(\mathbf{x}_1) - f^*)}{\sum_{t=1}^T \eta_t} + \frac{\sum_{i=1}^T \eta_t\xi_t}{\sum_{t=1}^T \eta_t} + \frac{L'\sum_{i=1}^T \eta_t\sqrt{\xi_t}}{\sum_{t=1}^T \eta_t},$$

*where $G(\mathbf{x}_t) = \frac{1}{\eta_t}(\mathbf{x}_t - \mathrm{proj}_{\mathcal{X}}(\mathbf{x}_t - \eta_t\mathbf{g}_t))$ is the gradient mapping.*

*Proof.* By $L$-smoothness, we have

$$f(\mathbf{x}_{t+1}) - f(\mathbf{x}_t) \leq \mathbf{g}_t^\top(\mathbf{x}_{t+1} - \mathbf{x}_t) + \frac{1}{2}L\|\mathbf{x}_{t+1} - \mathbf{x}_t\|^2$$
$$= -\eta_t\mathbf{g}_t^\top\widehat{G}(\mathbf{x}_t) + \frac{1}{2}L\eta_t^2\|\widehat{G}(\mathbf{x}_t)\|^2$$
$$\leq -\eta_t\mathbf{g}_t^\top\widehat{G}(\mathbf{x}_t) + \frac{1}{2}\eta_t\|\widehat{G}(\mathbf{x}_t)\|^2,$$

where the second line plugs in the update $\mathbf{x}_{t+1} = \mathbf{x}_t - \eta_t\widehat{G}(\mathbf{x}_t)$ and the third line is due to $\eta_t \leq \frac{1}{L}$. Now we analyze the two terms separately. For the first term, we have

$$-\eta_t\mathbf{g}_t^\top\widehat{G}(\mathbf{x}_t) = -\eta_t\mathbf{g}_t^\top\left(\widehat{G}(\mathbf{x}_t) - G(\mathbf{x}_t)\right) - \eta_t\mathbf{g}_t^\top G(\mathbf{x}_t)$$
$$\leq -\eta_t\mathbf{g}_t^\top\left(\widehat{G}(\mathbf{x}_t) - G(\mathbf{x}_t)\right) - \eta_t\|G(\mathbf{x}_t)\|^2,$$

where the second inequality uses Lemma 17. For the second term, we have

$$\frac{1}{2}\eta_t\|\widehat{G}(\mathbf{x}_t)\|^2 = \frac{1}{2}\eta_t\|\widehat{G}(\mathbf{x}_t) - G(\mathbf{x}_t) + G(\mathbf{x}_t)\|^2$$
$$= \frac{1}{2}\eta_t\|\widehat{G}(\mathbf{x}_t) - G(\mathbf{x}_t)\|^2 + \eta_t\left(\widehat{G}(\mathbf{x}_t) - G(\mathbf{x}_t)\right)^\top G(\mathbf{x}_t) + \frac{1}{2}\eta_t\|G(\mathbf{x}_t)\|^2$$
$$\leq \frac{1}{2}\eta_t\|\hat{\mathbf{g}}_t - \mathbf{g}_t\|^2 + \eta_t\left(\widehat{G}(\mathbf{x}_t) - G(\mathbf{x}_t)\right)^\top G(\mathbf{x}_t) + \frac{1}{2}\eta_t\|G(\mathbf{x}_t)\|^2$$

Summing them together, we have

$$f(\mathbf{x}_{t+1}) - f(\mathbf{x}_t) \leq \frac{1}{2}\eta_t\|\hat{\mathbf{g}}_t - \mathbf{g}_t\|^2 + \eta_t\left(\widehat{G}(\mathbf{x}_t) - G(\mathbf{x}_t)\right)^\top(G(\mathbf{x}_t) - \mathbf{g}_t) - \frac{1}{2}\eta_t\|G(\mathbf{x}_t)\|^2$$
$$\leq \frac{1}{2}\eta_t\|\hat{\mathbf{g}}_t - \mathbf{g}_t\|^2 + \eta_t\|\widehat{G}(\mathbf{x}_t) - G(\mathbf{x}_t)\| \cdot \|\mathbf{g}_t\| - \frac{1}{2}\eta_t\|G(\mathbf{x}_t)\|^2$$
$$\leq \frac{1}{2}\eta_t\xi_t + \eta_t L'\sqrt{\xi_t} - \frac{1}{2}\eta_t\|G(\mathbf{x}_t)\|^2,$$

where the second line uses Cauchy-Schwarz inequality.

A telescoping sum gives

$$\sum_{t=1}^{T} \eta_t \|G(\mathbf{x}_t)\|^2 \leq \sum_{t=1}^{T} \eta_t \xi_t + 2L' \sum_{t=1}^{T} \eta_t \sqrt{\xi_t} + 2(f(\mathbf{x}_1) - f(\mathbf{x}_{T+1})),$$

which results in

$$\min_{1 \leq t \leq T} \|G(\mathbf{x}_t)\|^2 \leq \frac{2(f(\mathbf{x}_1) - f^*)}{\sum_{t=1}^{T} \eta_t} + \frac{\sum_{i=1}^{T} \eta_t \xi_t}{\sum_{t=1}^{T} \eta_t} + \frac{L' \sum_{i=1}^{T} \eta_t \sqrt{\xi_t}}{\sum_{t=1}^{T} \eta_t}.$$

$\square$

The rest of this section proves all theorems and their corollaries in the main paper.

**Theorem 1.** *Let $f \in \mathcal{H}$ whose smoothness constant is L. Running Algorithm 1 with constant batch size $b_t = b$ and step size $\eta_t = \frac{1}{L}$ for T iterations outputs a sequence satisfying*

$$\min_{1 \leq t \leq T} \|\nabla f(\mathbf{x}_t)\|^2 \leq \frac{1}{T}\big(2L(f(\mathbf{x}_1) - f^*)\big) + B^2 \cdot E_{d,k,0}(b). \tag{5}$$

*Proof.* By Lemma 1 and Assumption 3, we can bound the bias in the iteration $t$ as

$$\|\nabla f(\mathbf{x}_t) - \nabla \mu_{\mathcal{D}_t}(\mathbf{x}_t)\|^2 \leq B^2 \operatorname{tr}\big(\nabla k_{\mathcal{D}_t}(\mathbf{x}_t, \mathbf{x}_t)\nabla^\top\big).$$

By Lemma 8, the trace in the RHS can be bounded by the error function $E_{d,k,\sigma}(b)$. Thus, the gradient bias is $B^2 E_{k,k,\sigma}(b)$. Applying Lemma 15 with $\eta_t = \frac{1}{L}$ and $\xi_t = B^2 E_{k,k,\sigma}(b)$ finishes the proof. $\square$

**Corollary 1.** *Under the same assumptions of Theorem 1, using batch size $b_t = d + 1$, we have*

$$\min_{1 \leq t \leq T} \|\nabla f(\mathbf{x}_t)\|^2 \leq \frac{1}{T}\big(2L(f(\mathbf{x}_1) - f^*)\big).$$

*Therefore, the total number of samples $n = \mathcal{O}(dT)$ and the squared gradient norm $\|\nabla f(\mathbf{x}_t)\|^2$ converges to zero at the rate $\mathcal{O}(d/n)$.*

*Proof.* By Lemma 2, we have $E_{d,k,\sigma}(d + 1) = 0$. Plugging it into Theorem 1 gives the rate in the iteration number $T$. To get the rate in samples $n$, note that $n = \sum_{t=1}^{T}(d + 1) = (d + 1)T$. Plugging $T = \frac{n}{d+1}$ into the rate finishes the proof. $\square$

**Theorem 2.** *For $0 < \delta < 1$, suppose $f$ is a GP sample whose smoothness constant is L w.p. at least $1 - \delta$. Algorithm 1 with batch size $b_t$ and step size $\eta_t = \frac{1}{L}$ produces a sequence satisfying*

$$\min_{1 \leq t \leq T} \|\nabla f(\mathbf{x}_t)\|^2 \leq \frac{1}{T}\big(2L(f(\mathbf{x}_1) - f^*)\big) + \frac{1}{T}\sum_{t=1}^{T} C_t E_{d,k,\sigma}(b_t) \tag{6}$$

*with probability at least $1 - 2\delta$, where $C_t = 2\log\big((\pi^2/6)(t^2/\delta)\big)$.*

*Proof.* By Lemma 11, we have

$$\|\nabla f(\mathbf{x}_t) - \nabla \mu_{\mathcal{D}_t}(\mathbf{x}_t)\|^2 \leq C_t \operatorname{tr}\big(\nabla k_{\mathcal{D}_t}(\mathbf{x}_t, \mathbf{x}_t)\nabla^\top\big)$$

with probability at least $1 - \delta$. The trace on the RHS can be further bounded by the error function $E_{d,k,\sigma}(b_t)$ by Lemma 8. Applying the union bound, with probability at least $1 - 2\delta$, the inequality

$$\|\nabla f(\mathbf{x}_t) - \nabla \mu_{\mathcal{D}_t}(\mathbf{x}_t)\|^2 \leq C_t E_{d,k,\sigma}(b_t)$$

holds for all $t \geq 0$ and $f$ is $L$-smooth. Applying Lemma 15 with $\eta_t = \frac{1}{L}$ and $\xi = C_t E_{d,k,\sigma}(b_t)$ finishes the proof. $\square$

**Corollary 2.** *Let $k(\cdot, \cdot)$ be either the RBF kernel or the $\nu = 2.5$ Matérn kernel. Under the same conditions as Theorem 2, if*

$$b_t = \begin{cases} d\log^2 t; \\ dt; \\ dt^2, \end{cases} \quad then \quad \min_{1 \leq t \leq T} \|\nabla f(\mathbf{x}_t)\|^2 = \begin{cases} \mathcal{O}(1/T) + \mathcal{O}(\sigma d); \\ \mathcal{O}(\sigma d T^{-\frac{1}{2}} \log T) = \mathcal{O}(\sigma d^{\frac{5}{4}} n^{-\frac{1}{4}} \log n); \\ \mathcal{O}(\sigma d T^{-1} \log^2 T) = \mathcal{O}(\sigma d^{\frac{4}{3}} n^{-\frac{1}{3}} \log^2 n), \end{cases}$$

*with probability at least $1 - 2\delta$. Here, $T$ is the total number of iterations and $n$ is the total number of samples queried.*

*Proof.* By Theorem 2, with probability at least $1 - 2\delta$, we have

$$\min_{1 \leq t \leq T} \|\nabla f(\mathbf{x}_t)\|^2 \leq \tfrac{1}{T}\big(2L(f(\mathbf{x}_1) - f^*)\big) + \tfrac{1}{T}\sum_{t=1}^{T} C_t E_{d,k,\sigma}(b_t)$$

The proof boils down to bounding the average cumulative bias. The full details is in Appendix D. $\square$

**Theorem 3.** *Under the same conditions as Corollary 2, without Assumption 2, using the projected update rule (7), Algorithm 1 obtains the following rates:*

$$if \quad b_t = \begin{cases} dt; \\ dt^2, \end{cases} \quad then \quad \min_{1 \leq t \leq T} \|G(\mathbf{x}_t)\|^2 = \begin{cases} \mathcal{O}(\sigma d^{\frac{5}{4}} n^{-\frac{1}{4}} \log n + \sigma^{\frac{1}{2}} d^{\frac{5}{8}} n^{-\frac{1}{8}} \log n); \\ \mathcal{O}(\sigma d^{\frac{4}{3}} n^{-\frac{1}{3}} \log^2 n + \sigma^{\frac{1}{2}} d^{\frac{2}{3}} n^{-\frac{1}{6}} \log n), \end{cases}$$

*with probability at least $1 - 2\delta$. Here, $n$ is the total number of samples queried.*

*Proof.* Since $f$ is twice differentiable on a compact set $\mathcal{X}$, its gradient norm $\|\nabla f(\mathbf{x})\|$ attains a maximum. Thus, there exists a constant $L'$ such that $L' \geq \|\nabla f(\mathbf{x})\|$ for all $\mathbf{x} \in \mathcal{X}$. By Lemma 18 and a similar argument in Theorem 2, with probability at least $1 - 2\delta$, we have

$$\min_{1 \leq t \leq T} \|G(\mathbf{x}_t)\|^2 \leq \tfrac{1}{T}\big(2L(f(\mathbf{x}_1) - f^*)\big) + \tfrac{1}{T}\sum_{t=1}^{T} C_t^{(2)} E_{d,k,\sigma}(b_t) + \tfrac{1}{T}\sum_{t=1}^{T} C_t^{(2)} \sqrt{E_{d,k,\sigma}(b_t)},$$

where $C_t^{(1)}$ and $C_t^{(2)}$ are constants growing in $\mathcal{O}(\log t)$. By Lemmas 4 and 5, plug in the error function $E_{d,k,\sigma}(b) = \mathcal{O}(\sigma d^{\frac{3}{2}} b^{-\frac{1}{2}})$. The rest of the proof follows a similar argument in Corollary 2, as shown in Appendix D. $\square$

Finally, we present a convergence result for GP sample path under noiseless assumption.

**Theorem 4.** *For $0 < \delta < 1$, suppose $f$ is a GP sample whose smoothness constant is $L$ with probability at least $1 - \delta$. Assuming $\sigma = 0$, Algorithm 1 with batch size $b_t = d + 1$ and step size $\eta_t = \frac{1}{L}$ produces a sequence satisfying*

$$\min_{1 \leq t \leq T} \|\nabla f(\mathbf{x}_t)\|^2 \leq \frac{2L(f(\mathbf{x}_1) - f^*)}{T}$$

*with probability at least $1 - 2\delta$.*

*Proof.* By Theorem 2, we have

$$\min_{1 \leq t \leq T} \|\nabla f(\mathbf{x}_t)\|^2 \leq \frac{2L(f(\mathbf{x}_1) - f^*)}{T} + \frac{1}{T}\sum_{t=1}^{T} C_t E_{d,k,\sigma}(d+1)$$

with probability at least $1 - 2\delta$. By Lemma 2, $E_{d,k,\sigma}(d+1) = 0$, the second term is essentially zero, which finishes the proof. $\square$

# D   Optimizing the Batch Size

In this section, we optimize the batch size in Theorem 2 and give explicit convergence rates. The discussion in this section will give a proof for Corollary 2.

For the RBF kernel and $\nu = 2.5$ Matérn kernel, by Theorem 2, Lemma 4 and Lemma 5, we have shown the following bound

$$\min_{1 \leq t \leq T} \|\nabla f(\mathbf{x}_t)\|^2 \lesssim \frac{1}{T} + \frac{1}{T} \sum_{t=1}^{T} E_{d,k,\sigma}(b_t) \log t$$

$$\lesssim \frac{1}{T} + \frac{\sigma d^{\frac{3}{2}}}{T} \sum_{t=1}^{T} b_t^{-\frac{1}{2}} \log t.$$

We discuss polynomially growing batch size $b_t = dt^a$, where $a > 0$. Then we have

$$\min_{1 \leq t \leq T} \|\nabla f(\mathbf{x}_t)\|^2 \lesssim \frac{1}{T} + \sigma d \cdot \frac{1}{T} \sum_{t=1}^{T} t^{-\frac{1}{2}a} \log t.$$

We discuss three cases: $0 < a < 2$, $a = 2$ and $a > 2$.

**Case 1.** When $0 \leq a < 2$, the infinite sum $\sum_{t=1}^{T} t^{-\frac{1}{2}a} \log t$ diverges. Its growth speed is on the order of $\mathcal{O}(T^{1-\frac{1}{2}a} \log T)$. The total number of samples $n = \sum_{t=1}^{T} b_t = \mathcal{O}(dT^{a+1})$, and thus $T = \mathcal{O}(d^{-\frac{1}{a+1}} n^{\frac{1}{a+1}})$. Thus, the rate is

$$\min_{1 \leq t \leq T} \|\nabla f(\mathbf{x}_t)\|^2 \lesssim T^{-1} + \sigma d \cdot T^{-\frac{1}{2}a} \log T$$

$$\lesssim d^{\frac{1}{a+1}} n^{-\frac{1}{a+1}} + \sigma d^{\frac{3a+2}{2(a+1)}} n^{-\frac{a}{2(a+1)}} \log n$$

$$\lesssim \sigma d^{\frac{3a+2}{2(a+1)}} n^{-\frac{a}{2(a+1)}} \log n,$$

where the last inequality uses the fact that the second term dominates the rate when $0 < a < 2$.

**Case 2.** When $a = 2$, the infinite sum $\sum_{t=1}^{T} t^{-1} \log t$ diverges. Its growth speed is on the order of $\mathcal{O}(\log^2 T)$. The total number of samples is $n = d \sum_{t=1}^{T} t^2 = \mathcal{O}(dT^3)$, and thus $T = \mathcal{O}(d^{-\frac{1}{3}} n^{\frac{1}{3}})$. Then the rate is

$$\min_{1 \leq t \leq T} \|\nabla f(\mathbf{x}_t)\|^2 \lesssim \frac{1}{T} + \frac{\sigma d \log^2 T}{T}$$

$$\lesssim \sigma d T^{-1} \log^2 T$$

$$\lesssim \sigma d^{\frac{4}{3}} n^{-\frac{1}{3}} \log^2 n.$$

**Case 3.** When $a > 2$, the infinite sum $\sum_{t=1}^{T} t^{-\frac{1}{2}a} \log t$ converges to a constant when $T \to \infty$. The total number of samples $n = \mathcal{O}(dT^{a+1})$. Thus, the rate is

$$\min_{1 \leq t \leq T} \|\nabla f(\mathbf{x}_t)\|^2 \lesssim T^{-1} + \sigma d T^{-1}$$

$$\lesssim \sigma d^{\frac{a+2}{a+1}} n^{-\frac{1}{a+1}}.$$

Note that $a = 2$ achieves the fastest rate $\mathcal{O}(n^{-\frac{1}{3}})$ in terms of samples $n$.

Now we discuss a batch size with logarithmic growth $b_t = d \log^2 t$. We have

$$\min_{1 \leq t \leq T} \|\nabla f(\mathbf{x}_t)\|^2 \lesssim \frac{1}{T} + \frac{\sigma d}{T} \sum_{t=1}^{T} \mathcal{O}(1)$$

$$\lesssim \frac{1}{T} + \sigma d.$$

# E    Additional Experiments

This section presents additional experimental details and additional numerical simulations.

**Details of Figure 2.**    In Figure 2a, we plot the error function starting from $b = 20$ to make sure $b \geq 2d$ so that Lemma 5 indeed applies. The decay rate $\mathcal{O}(\sigma d^{\frac{3}{2}} b^{-\frac{1}{2}})$ has a (leading) hidden constant of $\frac{15\sqrt{2}}{2}$ inside the big $\mathcal{O}$ notation (see the proof of Lemma 5), and thus the bounds plotted in Figure 2 are multiplied by this constant. Otherwise, the expression $\sigma d^{\frac{3}{2}} b^{-\frac{1}{2}}$ alone is not a valid upper bound.

**ReLU.**    The ReLU function $\max\{0, x\}$ is non-differentiable at $x = 0$. Nevertheless, thanks to convexity the subdifferential at $x = 0$ is defined as $[0, 1]$. We estimate the "derivative" at $x = 0$ by minimizing the acquisition function. The estimate $\mu'_{\mathcal{D}}(0)$ and queries are plotted in Figure 5. The estimated derivative $\mu'_{\mathcal{D}}(0)$ is always in $[0, 1]$. Thus, the posterior mean gradient $\mu'_{\mathcal{D}}(0)$ produces a subgradient in this case.

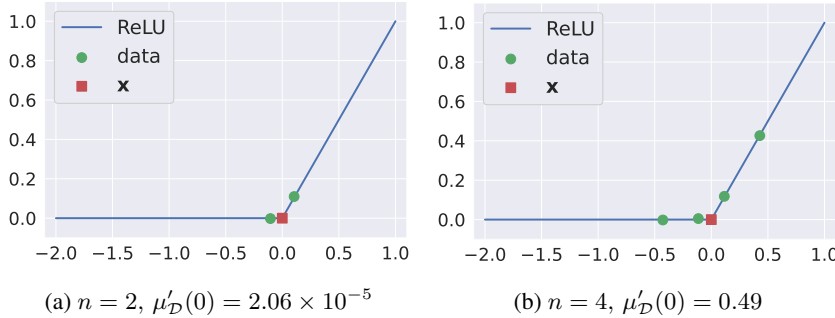

(a) $n = 2$, $\mu'_{\mathcal{D}}(0) = 2.06 \times 10^{-5}$      (b) $n = 4$, $\mu'_{\mathcal{D}}(0) = 0.49$

Figure 5: Estimating the "derivative" of ReLU at $x = 0$ with noisy observations ($\sigma = 0.01$).

