# OpenReview forum: "The Behavior and Convergence of Local Bayesian Optimization"
_NeurIPS.cc/2023/Conference — NeurIPS 2023 spotlight_

### Official Review · Reviewer_pxRe · 2023-07-03

**Soundness:** 3 good
**Presentation:** 3 good
**Contribution:** 3 good
**Rating:** 8
**Confidence:** 4

**Summary:**

Local Bayesian optimization has been popular high-dimensional optimization methods, which achieves the state-of-the-art empirical performance over various benchmarks. This paper investigates the behavior of local Bayesian optimization (BO) methods. Under a prototype algorithm, the authors empirically  demonstrate superior performance of the stationary points found by local BO compared to vanilla BO and grid search. Under the assumption that the function generating parameter are known, they theoretically and experimentally analyze the convergence rate of local BO methods.

**Strengths:**

1. I think the study problem in this paper is important and the theoretical experiment results partially explains the good performance of local BO.

2. The theoretical analysis is easy to follow, and gives insight of how to design local BO hyperparameters according to the problem dimension.

**Weaknesses:**

1. The prototype local BO algorithm samples points to reduce the gradient estimation unceratinty, and move search center according to the estimated mean gradient, which may not be representative of all local BO methods such as trust region Bayesian optimization (TuRBO) [1].

2. The assumptions in theoretical analysis may not hold in many real-world problems.

[1] Eriksson, David, et al. "Scalable global optimization via local Bayesian optimization." Advances in neural information processing systems 32 (2019).

**Questions:**

1. Many experiment in this paper use GP sample path as the objective function, with a dimension up to 100 (Figure 3). How to sample a function from GP under such high-dimensional input space? Any discretization is used during the function sampling?


**Limitations:**

The authors have addressed the limilations. I did not find potential negative societal impact of this paper.

---

> ### Author Rebuttal · Authors · 2023-08-07
>
> Thank you for your comments. We provide our response as follows.
>
> **Comment:** The assumptions in theoretical analysis may not hold in many real-world problems.
>
> We definitely agree that these assumptions may not hold in real-world problems – most real world functions are not GP sample paths. The assumptions we use (RKHS/GP sample) are the most commonly used assumptions in the BO theory literature because proving convergence in the black-box setting with arbitrary model misspecification is generally a pretty daunting task.
>
> With that said, interestingly, there actually is hope for relaxed assumptions for local BO. For local BO, we don’t need the GP to be a “potentially reasonable” surrogate model of the function everywhere in the input space. Intuitively, we only need sufficient smoothness to be able to estimate the gradient at the current iterate $x_t$ – all other areas of the input space are irrelevant at iteration t. It’s interesting future work to consider what the weakest assumptions are that enable provable GP-based gradient estimation.
>
> **Comment:** How to sample a function from GP under such a high-dimensional input space? Any discretization is used during the function sampling?
>
> We did not use discretization in our experiment -– see Line 99. We adapted the pathwise sampling technique of Wilson et al. (2021) as implemented in BoTorch, but using exact conditional calculations rather than variational / inducing point conditional calculations (since our goal was not improved asymptotic complexity).

---

> > ### Comment · Reviewer_pxRe · 2023-08-13
> >
> > Thanks for your clarification. I think this paper inspires me a lot.

---

### Official Review · Reviewer_GVja · 2023-07-05

**Soundness:** 4 excellent
**Presentation:** 4 excellent
**Contribution:** 4 excellent
**Rating:** 8
**Confidence:** 4

**Summary:**

Local Bayesian optimization methods are widely used to cope with the curse of dimensionality when optimizing high-dimensional black-box functions. Although these methods have very good performance empirically, virtually all theoretical results focus on the global optimization setting by proving bounds on the optimality gap. This paper shows that under typical assumptions a particular variant of local Bayesian optimization (GIBO) provably converges to a stationary point in polynomial time.

**Strengths:**

I greatly enjoyed this paper. It is well-written and easy to follow, and answers important questions with proof techniques that are novel to me. I am very happy to see that the authors included some empirical studies to verify the integrity of their results.

**Weaknesses:**

This work is ready for publication.

**Questions:**

- based on the description in line 113 it seems the x axes in Figure 1 are mislabeled, is that correct?

- I found the following statement on line 220 a bit vague: "However, because observations of f are made with noise, the noise inherit too [sic] both evaluations of f is amplified by 1/2h as h -> 0." Can you please elaborate?

- I found the second paragraph of the discussion intriguing. Do you think the noise in the BayesOpt gradient estimate may play a similar role as the minibatch gradient noise in SGD? My understanding of the argument is minibatch gradient noise tends to cause SGD to find wider basins in the loss surface, which some have argued is linked to better generalization. In this context we aren't concerned with generalization per se, but perhaps one could argue that wider basins of f are preferable since they will be more forgiving if the estimate of the stationary point is slightly off.

- in practice it is quite common to estimate the GP surrogate hyperparameters through the log marginal likelihood, can you comment on how this might affect your analysis and empirical behavior?

- how might one start thinking about "convergence" in the multi-objective case? Could we prove that local BayesOpt converges to a locally non-dominated point?

- although this paper did not propose GIBO, can you comment on its numerical stability in practice? In my experience look-ahead acquisition functions tend to exhibit poor numerical behavior, which I imagine is only amplified by introducing an additional derivative operator.

**Limitations:**

I see two primary limitations

- the analysis is restricted to GIBO. It would be interesting to know if any similar arguments can be made for TurBO [1], which is quite widely used.

- the analysis relies on fairly strong (albeit common) assumptions on the nature of f and the specification of the surrogate. It seems plausible to me that (for example) BayesOpt could find a stationary point even if the surrogate is misspecified, assuming f is not too pathological.

[1] Eriksson, David, et al. "Scalable global optimization via local Bayesian optimization." Advances in neural information processing systems 32 (2019).

---

> ### Author Rebuttal · Authors · 2023-08-07
>
> Thank you for your comments. We provide our response as follows.
>
> **Comment:** The x-axis of Figure 1.
>
> Thanks for spotting this typo! Indeed, the x-axis in Figure 1 is the indices of the dimension list {1, 5, 10, 20, 30, 50}. We will replot the x-axis of the figure.
>
> **Comment:** Line 220.
>
> We will rewrite this sentence in the revision. The central differencing scheme has $O(h^2)$ truncation error, and thus $\frac{\partial f}{\partial x} = \frac{1}{2h} \big( f(x + h) - f(x - h)\big) + O(h^2)$. To reduce the estimation error, we need a small $h$. However, the function evaluation is corrupted by a Gaussian in the presence of noise. The right hand side becomes $\frac{1}{2h} \big( f(x + h) - f(x - h) + 2 \epsilon\big) + O(h^2)$, where $\epsilon$ is Gaussian ($2\epsilon$ is due to two noisy function evaluations). When $h \to 0$, the denominator $\frac{1}{2h}$ blows up the noise. This illustrates that noise creates a sort of general hardness in estimating gradients, whether you’re using a GP or not.
>
> **Comment:** Second paragraph of discussion. Do you think the noise in the BayesOpt gradient estimate may play a similar role as the mini batch gradient noise in SGD?
>
> That’s an interesting question. We think the answer might be yes potentially. Besides the case you mentioned, there is another intuitive case. Due to the estimation error, local BO may be less likely to be trapped by saddle points (a particular type of bad stationary points). The intuition is that convergence to saddle points is unstable — a slight perturbation may let the algorithm find a decent direction and therefore escape the saddle point.
>
> Here we use the term *algorithmic bias* to broadly refer to this phenomenon: local BO algorithms (e.g. GIBO), are able to find good solutions (which is demonstrated by their strong empirical performance in previous research), despite the existence of potential bad local solutions/stationary points. It is not completely well-defined unfortunately but may be worth studying in the future.
>
> **Comment:** Estimating the hyperparameters through maximizing log likelihood.
>
> Our theoretical analysis assumes a kernel with fixed hyperparameters, which is relatively standard in the theoretical BO literature. Extending to unknown hyperparameters is an interesting future work. In practical settings, we should of course generally expect that estimating the hyperparameters improves the model fit, which typically in turn improves the convergence. The empirical performance of local BO methods in settings where hyperparameters are learned is of course well established in prior work.
>
> **Comment:** Convergence in the multi-objective case.
>
> It might be possible to extend our results in multi-objective settings. One of the intermediate steps in our proof is to show that the objective decreases in every iteration if the gradient estimation error is small (see the equations below Line 395). If we can show a similar property holds in the multi-objective setting (e.g., each iteration decreases all objectives), then it might be possible to prove convergence. Note that decreasing objectives locally is a very natural requirement for local BO algorithms. We think it is interesting to check if certain multi-objective BO algorithms satisfy this property (or propose new algorithms satisfying this property).
>
> **Comment:** Numerical stability of GIBO.
>
> Optimizing the acquisition function of GIBO is generally stable with good initialization. The resulting designs typically center around the current iterate $\mathbf{x}$ (e.g. Figure 4). In practice, however, updating the iterates $\mathbf{x}_{t + 1} = \mathbf{x}_t - \eta \mathbf{g}_t$ may require careful attention. The estimated gradient $\mathbf{g}_t$ is only an **approximation** of the true gradient. While in theory the approximation error can be bounded by our theorems, in practice the approximation error is affected by two additional complications: a) model misspecification and b) small batch size in the noisy setting. Thus, it is recommended to use small step sizes to avoid overshooting, or even line search strategies. Oftentimes normalizing the estimated gradient $\mathbf{g}_t$ is helpful (Muller et al., 2021).

---

> > ### Comment · Reviewer_GVja · 2023-08-11
> > **Acknowledgement**
> >
> > Thanks for your response, this is great work and I remain strongly in favor of acceptance.

---

### Official Review · Reviewer_kgRX · 2023-07-08

**Soundness:** 3 good
**Presentation:** 3 good
**Contribution:** 3 good
**Rating:** 6
**Confidence:** 3

**Summary:**

The paper investigates the behavior and convergence property of the local Bayesian optimization approach, in particular, the method GIBO [16]. The paper provides the convergence rates of GIBO in both the noiseless and noisy settings. The paper also performs various empirical experiments to understand the tightness of the derived bounds. It also conducts some experiments to understand some other properties of local BO like multiple restarts and when the objective function is non-differentiable.

**Strengths:**

+ The paper tackles an important and difficult problem, which is to perform theoretical analysis for the local BO approach
+ The paper writing is very clear and easy to understand. The paper is really technical; it has a lot of theory, but thanks to the clear writing, all the theoretical analysis is much easier to understand.
+ The theoretical analysis on the convergence rates of GIBO seems to be sound to me (based on my overall understanding regarding the theoretical analysis of BO – I couldn’t dig into the proof in a very detailed manner).
+ Some empirical experiments are also conducted to understand the tightness of the derived bounds.


**Weaknesses:**

+ The paper actually focuses only on one of the local BO approaches (the GIBO method) and this approach is actually quite different compared to some other popular local BO approaches like TurBO. The GIBO approach is based on gradient descent, but the other local BO approach like TurBO is still based on BO. The paper should make very clear regarding the focus of the theoretical analysis.
+ Another weakness of the paper is that it still relies on the assumption that are used to analyze the global BO including the assumption that the objective function is a sample from a GP with known hyperparameters. Actually one of the main motivation of the local BO approach is based on the observation that the objective function can’t be model using one single global GP in the whole search domain. Therefore, the assumption of the theoretical analysis in this work seems to be quite strict, and therefore, the theoretical analysis (the bounds) may not shed too much light on the performance of the local BO approach.


**Questions:**

Please answer my comments in the Weaknesses section, and the following additional questions:
+ In the experiments (Section 6.1), how are the hyperparameters of the bounds or the algorithms set (like \beta) ?


**Limitations:**

The paper seems to not have a dedicated section describing the limitations of the work presented in the paper.

---

> ### Author Rebuttal · Authors · 2023-08-07
>
> Thank you for your comments. We provide our response as follows.
>
> **Comment:** The paper actually focuses only on one of the local BO approaches (the GIBO method).
>
> See the global response above.
>
> **Comment:** Sample path assumption with known hyperparameter.
>
> While this assumption is standard and widely used in the theoretical BO literature, there is also potential hope for relaxed assumptions for local BO. For local BO, we don’t need the GP to be a “potentially reasonable” surrogate model of the function everywhere in the input space. Intuitively, we only need sufficient smoothness to be able to estimate the gradient at the current iterate $x_t$ – all other areas of the input space are irrelevant at iteration t. It’s interesting future work to consider what the weakest assumptions are that enable provable GP-based gradient estimation. We plan to explore this possibility further in future work.
>
>
> Nevertheless, we believe that our results already reveal some advantages of local BO algorithms. Our convergence rates (for local stationary points) are **polynomial** both in the dimension $d$ and the number of samples $n$. This is in sharp contrast to any global BO algorithm, e.g. GP-UCB. In this sense, our results provide partial justifications for local BO algorithms and their fast convergence in practice (at least for a particular algorithm, GIBO).
>
> **Comment:** In the experiments (Section 6.1), how are the hyperparameters of the bounds or the algorithms set (like \beta) ?
>
> Figure 2 studies the tightness of the bound on the error function (Definition 2), which depends only on the kernel and not the objective function, and therefore makes no use of $\beta$. The kernel used in Figure 2 is a RBF kernel with a unit length scale, as in Theorem 3. The purpose of Figure 2 is to compare how quickly the posterior derivative variance can be decreased empirically to our theoretical upper bound on that rate. Since the posterior variance doesn’t depend on the actual labels collected (with fixed hyperparameters), and since the error function is measured at the origin with only the collected batch of data, the results in Figure 2 are independent of collected labels or any particular objective function.

---

> > ### Comment · Reviewer_kgRX · 2023-08-17
> > **Thank you for your response**
> >
> > Dear authors,
> >
> > Thank you for your response. The reponse has addressed some of my concerns, however, I still think the assumption regarding the objective function being a sample from a GP with known hyperparameters is still quite strong when analyzing the local BO approach, so I still keep my current rating.

---

### Official Review · Reviewer_aL9q · 2023-07-25

**Soundness:** 3 good
**Presentation:** 3 good
**Contribution:** 3 good
**Rating:** 8
**Confidence:** 3

**Summary:**

This paper gives theoretical considerations on local Bayesian optimization methods for black box optimization. In particular, for gradient-based local BO, when the objective function satisfies the smoothness assumption, the rate of convergence to the stationary point is derived for the noiseless and noisy cases, respectively.
In the noiseless case, local solutions can be found at a much faster rate with respect to the input dimension d than with conventional global BO.
Although the convergence rate is worse in the presence of observation noise than in the noiseless case, it returns to the noiseless rate in the limit of zero noise variance, suggesting that it is a reasonable bound.
The tightness of the derived convergence rates is validated by numerical experiments.


**Strengths:**

- The rate of convergence of the gradient-based local BO algorithm to the local optimal solution is theoretically derived, and the rate is expected to be somewhat tight (empirically validated).

- It is clear how the observational noise affects convergence in the local BO. Furthermore, by taking the limit of the noise variance parameter, the convergence rate connects the noiseless and noisy cases.

- The convergence rate results for the noiseless case are given for both cases where the objective function is an element of the RKHS (in the main text) and a sample path of a Gaussian process (in the appendix).


**Weaknesses:**

- The local BO discussed here is a gradient method type algorithm called GIBO in particular, and approaches that construct the GP itself locally, such as TurBO [Eriksson+ 2019], have not been considered.

- It would be nice to have a theory that connects convergence to a stationary point and regret analysis. In Section 3, the results suggest that local solutions are good enough compared to the global solution, and if this is a property that holds to some extent universally, then it seems reasonable to expect that the regrets of local search also have good properties (e.g., sublinear properties).

**Questions:**

- I think one advantage of local search is that it is compatible with parallel distributed processing (such as TurBO). Is it possible to extend the gradient-based local BO treated in this paper to parallel distributed processing as well? And if so, would some theory of  convergence for such methods be obtained by a direct extension of the results derived in this paper?

- Theorem 3 derives the upper bound of the error function for the squared exponential kernel. Can we easily obtain the upper bound of the error function for other kernels (e.g. Matern kernel)?

- In Lemma 2 and line 226, is $E_{d, k, \sigma}(b_t, d)$ mistake for $E_{d, k, \sigma}(b_t)$?

**Limitations:**

The authors appear to have adequately addressed the limitations of the study and, where applicable, the potential adverse social consequences.

---

> ### Author Rebuttal · Authors · 2023-08-07
>
> Thank you for your comments. We provide our response as follows.
>
> **Comment:** It would be nice to have a theory that connects convergence to a stationary point and regret analysis.
>
> We agree this is an interesting question to convert our convergence rates to global regret bounds. However, this is actually a fairly challenging problem that would probably constitute a large technical lift in its own right. We’d need to analyze the regret of finding a sequence of stationary points using random restarts. Essentially, this is a question about the relationship between stationary points of GP sample paths and the global optimum. To the best of our knowledge, there is little research on this question, although some of the theories by Adler (1981) may be relevant. We think this is interesting future work.
>
> **Comment:** Is it possible to extend the gradient-based local BO treated in this paper to parallel distributed processing as well?
>
> First we note that, while it’s definitely not obvious, search direction based BO methods actually do effectively use batch evaluation already! Most of the evaluation budget of methods like GIBO and MPD is spent acquiring batches of data Z to learn about the gradient. Optimizing the acquisition function for Z can and often is done for relatively large (tens to hundreds) batches of points at a time. Thus, even the sequential single restart version of GIBO makes reasonably effective use of parallel evaluation capabilities. If additional parallelization is required, considering multiple restarts simultaneously is possible.
>
> **Comment:** Upper bound of the error function for other kernels.
>
> We do now have an additional result on the Matern kernel ($\nu = 2.5$). Interestingly, we are able to prove the Matern kernel has the same rate as the RBF kernel, i.e., $E_{d, k, \sigma}(b) = \mathcal{O}(\sigma d^{\frac32}b^{-\frac12})$. This new proof slightly adapts the techniques in the paper. The main new technique is to use a rational function upper bound on the posterior covariance trace for the Matern kernel, which allows us to analyze the error function by minimizing this rational function upper bound.
>
> We conjecture that our proof can be generalized to the **entire** Matern family (widely used in practice) and the rational quadratic kernel. Additionally, we remark that the error function we proposed in this paper is a quantity analogous to the information gain in global BO analysis — bounds on the error function for **any** kernels immediately result in convergence rates by our proof.
>
> **Comment:** In Lemma 2 and Line 226…
>
> Correct. Indeed, the dependency on the dimension is through the subscript, not the argument. Thanks for spotting this typo.

---

> > ### Comment · Reviewer_aL9q · 2023-08-12
> > **Response to the authors' comments**
> >
> > Thank you very much for your detailed answers to my questions.
> > I guess I did not understand enough about parallel and distributed processing, and the authors' comments convinced me.
> > It is also very interesting to see the possibility of extending the error function bounds to important types of kernels. If this conjecture is true, the results of this paper would be even stronger.
> > Taking into account other review comments and responses, I will raise the score by one.

---

### Author Rebuttal · Authors · 2023-08-07

We thank all reviewers for their encouraging comments and helpful feedback on our submission.

All reviewers share a common question of whether and how our theory might be extended beyond descent direction approaches like GIBO to trust region algorithms like TuRBO. In general we agree it would be very interesting to analyze TuRBO theoretically, and we’re interested in investigating this as well. While at this point we think it’s probably possible, the analysis is slightly more involved than GIBO. In particular, we believe TuRBO must be more significantly modified than GIBO in order to analyze it. In particular, the rules for shrinking and expanding the trust region will likely be different than simply “grow after k successes, shrink after k failures.”

We will revise relevant sentences in the paper to make it clear that we have only proved the convergence rate for search direction style local BO methods and list the convergence of TuRBO as important future work.

---

### Decision · Program_Chairs · 2023-09-21

**Decision:**

Accept (spotlight)

**Comment:**

This paper investigates a theoretical behavior of the local Bayesian optimization, presenting its convergence rate for both noiseless and noisy cases. All reviewers agree that the paper has a solid contribution, touching an important and timely subject in BO. The authors did a good job in the author response. I believe that the paper is ready for publication.